# Measuring ligand-cell surface receptor affinities with axial line-scanning fluorescence correlation spectroscopy

**Antonia Franziska Eckert[1†§], Peng Gao[1,2†§‡], Janine Wesslowski[3†§], Xianxian Wang[3], Jasmijn Rath[1], Karin Nienhaus[1], Gary Davidson[3]\*, Gerd Ulrich Nienhaus[1,2,3,4]\***

[1]Institute of Applied Physics, Karlsruhe Institute of Technology, Karlsruhe, Germany; [2]Institute of Nanotechnology, Karlsruhe Institute of Technology, Karlsruhe, Germany; [3]Institute of Biological and Chemical Systems, Karlsruhe Institute of Technology, Karlsruhe, Germany; [4]Department of Physics, University of Illinois at Urbana-Champaign, Urbana, United States

**Abstract** Development and homeostasis of multicellular organisms is largely controlled by complex cell-cell signaling networks that rely on specific binding of secreted ligands to cell surface receptors. The Wnt signaling network, as an example, involves multiple ligands and receptors to elicit specific cellular responses. To understand the mechanisms of such a network, ligand-receptor interactions should be characterized quantitatively, ideally in live cells or tissues. Such measurements are possible using fluorescence microscopy yet challenging due to sample movement, low signal-to-background ratio and photobleaching. Here, we present a robust approach based on fluorescence correlation spectroscopy with ultra-high speed axial line scanning, yielding precise equilibrium dissociation coefficients of interactions in the Wnt signaling pathway. Using CRISPR/Cas9 editing to endogenously tag receptors with fluorescent proteins, we demonstrate that the method delivers precise results even with low, near-native amounts of receptors.

**\*For correspondence:**
gary.davidson@kit.edu (GD);
uli@uiuc.edu (GUN)

[†]These authors contributed equally to this work
[§]These authors are listed in alphabetical order

**Present address:** [‡]School of Physics and Optoelectronic Engineering, Xidian University, Xi'an, China

**Competing interests:** The authors declare that no competing interests exist.

## Introduction

In multicellular organisms, cell-cell communication is mediated to a large extent by cell surface receptors in the plasma membrane. They bind their cognate ligands from the extracellular space, which triggers a cascade of intracellular processes. Major signaling pathways form tightly regulated networks comprised of multiple receptors and protein ligands. Among these, Wnt signaling is an ancient and evolutionarily conserved, complex network that controls a variety of physiological processes essential for embryonic development and tissue homeostasis (*Clevers and Nusse, 2012*; *Niehrs, 2012*). There are 10 different Frizzled G-protein-coupled receptors binding 19 different Wnt glycolipoprotein ligands (in humans), and additional co-receptors and effector proteins are also crucially involved in the modulation of signal transduction (*Niehrs, 2012*). Knowledge about the strengths of the various biomolecular interactions is a prerequisite to understanding the function of such a complex network. Accordingly, the equilibrium dissociation coefficient, $K_D$, which determines the fraction of ligand-bound receptors at a particular ligand concentration, should be measured for all relevant interactions in the network. To this end, a multitude of biochemical and biophysical in vitro assays have been devised. Those assays, however, may not completely emulate the complex environment of living cells and tissues; consequently, researchers are striving to measure biomolecular interactions as closely as possible to the real-life situation (*Ries et al., 2009b*; *Ng et al., 2016*).

Fluorescence correlation spectroscopy (FCS) is a powerful method for measuring ligand-receptor interactions within the plasma membrane of live cells (*Bacia et al., 2006*; *Ries et al., 2009b*;

*Dörlich et al., 2015*). The technique requires that ligands and receptors are specifically labeled with different fluorescent markers so that their emission can be detected separately in two color channels. A confocal microscope allows a tiny observation volume (~1 fL) to be positioned within the sample, and the emitted fluorescence light is captured as a function of time (*Zemanová et al., 2003*). Fluctuations of the fluorescence intensity due to molecular diffusion into and out of the observation volume are analyzed by calculating intensity pair correlation functions, from which diffusion coefficients, related to the size of the diffusing entities, as well as local concentrations can be extracted. Importantly, dual-color cross-correlation analysis provides a means of selectively measuring the concentration of ligand-receptor complexes. Finally, $K_D$ values can be calculated from the local concentrations of free ligand as well as ligand-bound and ligand-free receptors.

The intrinsic dynamics of live cells and tissues poses considerable challenges to FCS measurements. Movement of the cell membrane within the observation volume gives rise to strong intensity fluctuations obscuring those resulting from molecular diffusion in the plane of the membrane. Line-scanning FCS (lsFCS) solves this problem by quickly scanning the observation volume perpendicularly through the membrane (*Ries et al., 2009a*). Another difficulty comes from the labeling procedure usually employed for live samples, that is the expression of fusion constructs of the proteins of interest with genetically encoded marker proteins of the GFP family (*Nienhaus and Nienhaus, 2014*). These markers are not as bright as synthetic dyes, and the signal-to-background ratio (SBR) is often small in the presence of the unavoidable cellular autofluorescence and especially at low expression levels, which are, however, desirable to avoid overexpression artifacts. The poor photostability of GFP-like proteins poses yet another challenge because the focused laser spot causes local photobleaching. Labels cannot be replenished efficiently from neighboring regions of the plasma membrane due to the low diffusivity of integral membrane proteins within biological membranes. Therefore, careful corrections are necessary to account for the loss of fluorescent labels.

Here, we present a novel framework for FCS-based measurement of ligand-receptor interactions featuring high sensitivity, precision and robustness. In contrast to earlier approaches (*Ries et al., 2009a*; *Ries et al., 2009b*; *Dörlich et al., 2015*), we employ axial (i.e. along the optical axis) line scanning at ultra-high speed by using a tunable acoustic gradient index of refraction (TAG) lens (*Duocastella et al., 2014*). This device allows us to sweep the observation volume through the membrane within a few microseconds and thus more than two orders of magnitude faster than with lateral line scanning by means of galvanometric scanners. With axial lsFCS, the correlation decay is significantly faster than with lateral lsFCS, resulting in shorter overall measurement times to reach comparable precision. Thus, changes in cellular morphology that may compromise the data are less likely to occur. Moreover, the much higher scanning frequency of TAG lens-based axial lsFCS results in an enhanced time resolution of the correlation functions. In dual-color cross-correlation experiments, optimal overlap of the observation volumes is much easier to achieve with axial lsFCS. It is also worth mentioning that, in a typical sample consisting of a layer of cells on a coverslip surface, axial scanning offers abundant spots for lsFCS measurements, greatly facilitating data collection. A new and powerful data analysis pipeline that includes background and photobleaching corrections allows us to extract precise equilibrium dissociation coefficients.

We have evaluated the method by measuring the binding of the Wnt signaling inhibitors human Dickkopf1 (DKK1) and Dickkopf2 (DKK2) to the Wnt co-receptor lipoprotein-receptor related protein 6 (LRP6) in human lung cancer (NCI-H1703) and human embryonic kidney (HEK293T) cells (*Mao et al., 2001*; *Cruciat and Niehrs, 2013*). For these interactions, $K_D$ values in the range of 0.3–0.5 nM (without error analysis) were determined by in vitro studies (*Bafico et al., 2001*; *Mao et al., 2001*; *Semënov et al., 2001*); a more recent paper reported $3 \pm 1$ nM (*Bourhis et al., 2010*).

As many biological studies involve overexpression of receptors far above the natural level, the effect of receptor density on the binding affinity is an issue of broad interest. To explore this issue, we varied the receptor density in the plasma membrane in the range 20–1000 $\mu m^{-2}$ by using three different ways to express LRP6 receptors fused to fluorescent proteins in the cells. The respective gene was (1) introduced via transient transfection, or (2) stably integrated into the genome of cells by stable transfection, or (3) integrated into a cell line at the endogenous LRP6 gene locus by CRISPR/Cas9 editing. We have observed an impressive consistency in the $K_D$ determinations using LRP6 fused to mCherry and tdTomato markers, which differ considerably in size, brightness and emission spectrum (*Shaner et al., 2005*), attesting to the accuracy of the results. As another example, we have studied the binding of DKK1 to Kremen2, a receptor modulating the canonical Wnt signaling pathway, and the impact of LRP6 on this interaction (*Davidson et al., 2002*; *Mao et al., 2002*).

## Results

### 3D confocal microscope with fast axial scanning

Our home-built microscope is shown schematically in *Figure 1a*; details of the design are provided in Materials and methods and *Figure 1—figure supplement 1*. For dual-color excitation, we select two out of three available lasers (blue (470 nm), green (561 nm) and red (640 nm)), pulsing every 25 ns with a mutual delay of 12.5 ns. This pulsed interleaved excitation scheme (*Müller et al., 2005*) yields two time-separated intervals of 12.5 ns during which photons are collected in two color channels; channel crosstalk is thus strongly reduced. Photons are registered by a time-correlated single photon counting (TCSPC) unit. For each photon, we record the absolute time from the start of data acquisition, that is the macro time, $t_M$, and the time elapsed since the previous laser pulse, that is the micro time, $t_\mu$. The observation volume, formed by tight focusing of the laser beam with a high-NA objective and a confocal detection pinhole, is scanned laterally (within the focal plane) by galvanometric mirrors. In addition, we use a TAG lens driven in resonance (147 kHz) to move the observation volume up and down along the optical axis by several micrometer in an oscillatory manner with a period of ~6.8 µs (*Figure 1—figure supplement 2*). In addition to $t_M$ and $t_\mu$, the TCSPC card stores the lateral positions, $x$ and $y$, of the galvanometer mirrors for each photon counting event as well as the timing pulse, $t_{TAG}$, at the beginning of each TAG lens oscillation cycle. From $t_{TAG}$, the $z$ position is determined by a mapping procedure accounting for the nonlinear axial displacement with time (*Figure 1—figure supplement 3*). As a consequence, the pixel dwell times vary with the displacement, so that the collected photon events need to be rescaled. Finally, dual-color 3D images can be reconstructed from the arrival times and locations of origin of all photons registered (*Figure 1*). Because axial scanning is fast, the voxel dwell times are short but can be effectively increased by multiple axial scans in succession at a chosen *x-y* position.

### Comparison of axial and lateral lsFCS

We compared the two scanning modes with Atto647N-labeled giant unilamellar vesicles (GUVs) as a simple model system offering excellent SBR (*Figure 2a*). We took axial lsFCS data by continuously scanning through the top of a GUV for 60 s in an oscillatory fashion (*Figure 2b*). These data can be visualized as kymographs, displaying the intensity during the axial sweeps along the *x*-axis and multiple sweeps in succession along the *y*-axis (*Figure 2c*). We determine the intensity as a function of time by integrating over the emission signal during each membrane crossing event. From the intensity time trace, we calculate an autocorrelation function, which can be fitted very well with a model function based on a circular observation area in the *x-y* plane and 2D diffusion within the membrane (*Figure 2d*),

$$G(\tau) = N^{-1}(1 + \tau/\tau_D)^{-1}. \tag{1}$$

Here, $\tau$ is the lag time and $\tau_D$ is the diffusional correlation time. The amplitude of the autocorrelation function, $G(0)$, equals the inverse of the average number of fluorescently labeled molecules in the observation area, $N$, which is the product of the average concentration (i.e. area density), $C$, and focal area, $A$, so that $N = C \cdot A = C \cdot \pi\omega_0^2$. Here, $\omega_0$ is the radial distance over which the intensity in the focal plane decays by a factor of $e^{-2}$ (for details, see *Figure 2—figure supplement 1* and Materials and methods). For lateral line scanning of the focus along the *x* direction, the resulting $G(\tau)$ has a much lower time resolution (millisecond scale, *Figure 2d*) due to the lower speed of galvanometer scanning. Moreover, $G(\tau)$ decays more slowly than with axial scanning because the observation area in the *y-z* plane, $A = \pi\omega_0 z_0$, is significantly larger and elongated (*Figure 2d*, insets) because the axial extension, $z_0$, of a confocal spot is about five times greater than the lateral one. Thus, there are more fluorophores in the observation area on average, resulting in a smaller amplitude, $G(0)$, of the autocorrelation function (*Figure 2d*). Fluorophores reside longer in the observation area and are thus more prone to photobleaching. The corresponding diffusional autocorrelation function,

$$G(\tau) = N^{-1}(1 + \tau/\tau_D)^{-1/2}(1 + S^2\tau/\tau_D)^{-1/2}, \tag{2}$$

contains an extra anisotropy parameter, $S = \omega_0/z_0$, which makes fitting more ambiguous. To conclude, axial lsFCS has several advantages over lateral scanning, resulting in a markedly reduced data acquisition time for comparable data quality (*Figure 2e*). Notably, an overall shorter measurement is

less likely to be spoiled by sample movement, which is unavoidable in experiments on live specimens.

## Analysis of fast axial lsFCS on live cells

In live-cell experiments, we survey cultured cells by collecting 3D image stacks to select suitable locations for axial lsFCS (*Figure 3a, b*). Subsequently, fluorescence intensity time traces are recorded from these spots for typically 60 s in two color channels. Notably, the unavoidable intrinsic

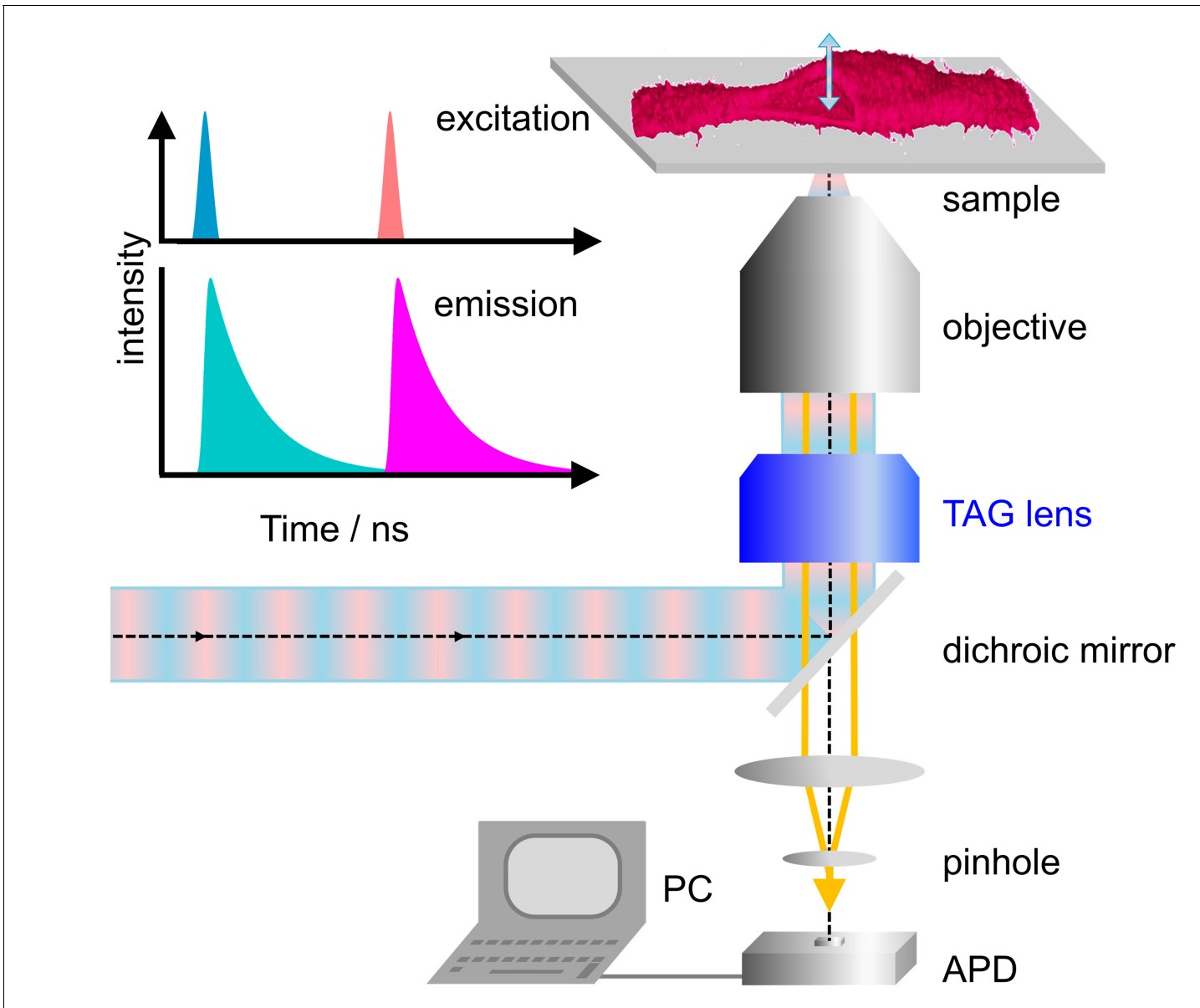

**Figure 1.** Dual-color confocal microscopy with pulsed interleaved excitation and ultrafast axial scanning with a tunable acoustic gradient index of refraction (TAG) lens. Shown are a schematic depiction of the microscope and (upper left) the excitation pulse sequence and the ensuing fluorescence emission. The sample cell (shown as a 3D image) is cut open to visualize axial scanning across the top membrane. The 3D image has been merged from four image slices (80 × 80 μm², 256 × 256 pixels, three scans, each with pixel dwell time 60 μs). APD, avalanche photodiode.

The online version of this article includes the following figure supplement(s) for figure 1:

**Figure supplement 1.** Schematic of the confocal microscope with fast axial scanning.

**Figure supplement 2.** Characterization of the confocal spot upon TAG lens scanning.

**Figure supplement 3.** Compensation of the nonlinear axial scanning by the TAG lens.

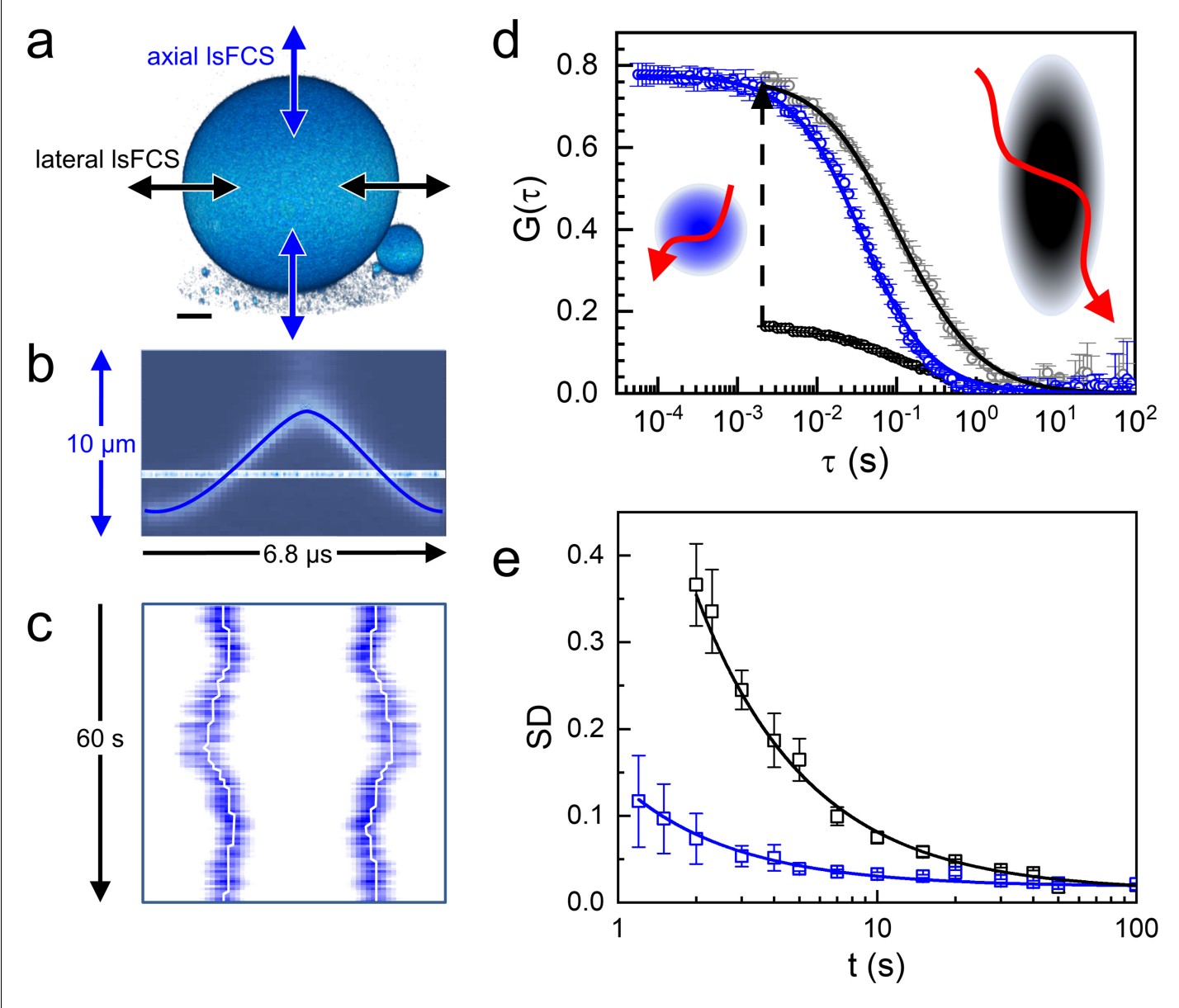

**Figure 2.** Axial and lateral line-scanning FCS on giant unilamellar vesicles (GUVs) labeled with lipid-bound Atto647N. (**a**) 3D image of a GUV (diameter 60 µm), showing the scanning directions of the focus across the membrane in lateral and axial lsFCS; scale bar, 10 µm. (**b**) Axial focus position during a single oscillation cycle of 6.8 µs. (**c**) Kymograph, horizontal axis: intensity during axial focus oscillations (averaged over 500 ms), vertical axis: sequence of 500 ms periods during 60 s of axial scanning. Lines mark the membrane positions. (**d**) Autocorrelation curves for axial (blue) and lateral (black) scanning; points: experimental data; lines: fits with *Equations (1) and (2)*, yielding $\tau_D = 36 \pm 1$ ms with axial scanning and $\tau_D = 46 \pm 2$ ms, $S = 4.0 \pm 0.3$ with lateral scanning, respectively. The lateral lsFCS autocorrelation data have also been scaled to the axial ones (grey) for better comparison. Observation areas on the plasma membrane are depicted schematically in the corresponding colors, possible dye trajectories are shown in red. (**e**) Standard deviation (SD), calculated for two groups of seven normalized lateral (black) and axial (blue) lsFCS curves, plotted as a function of the overall time of data acquisition. For details, see Materials and methods.

The online version of this article includes the following source data and figure supplement(s) for figure 2:

**Source data 1.** Numerical data of the plots in *Figure 2d,e*.
**Figure supplement 1.** Focus size determination by imaging multi-color fluorescent beads.

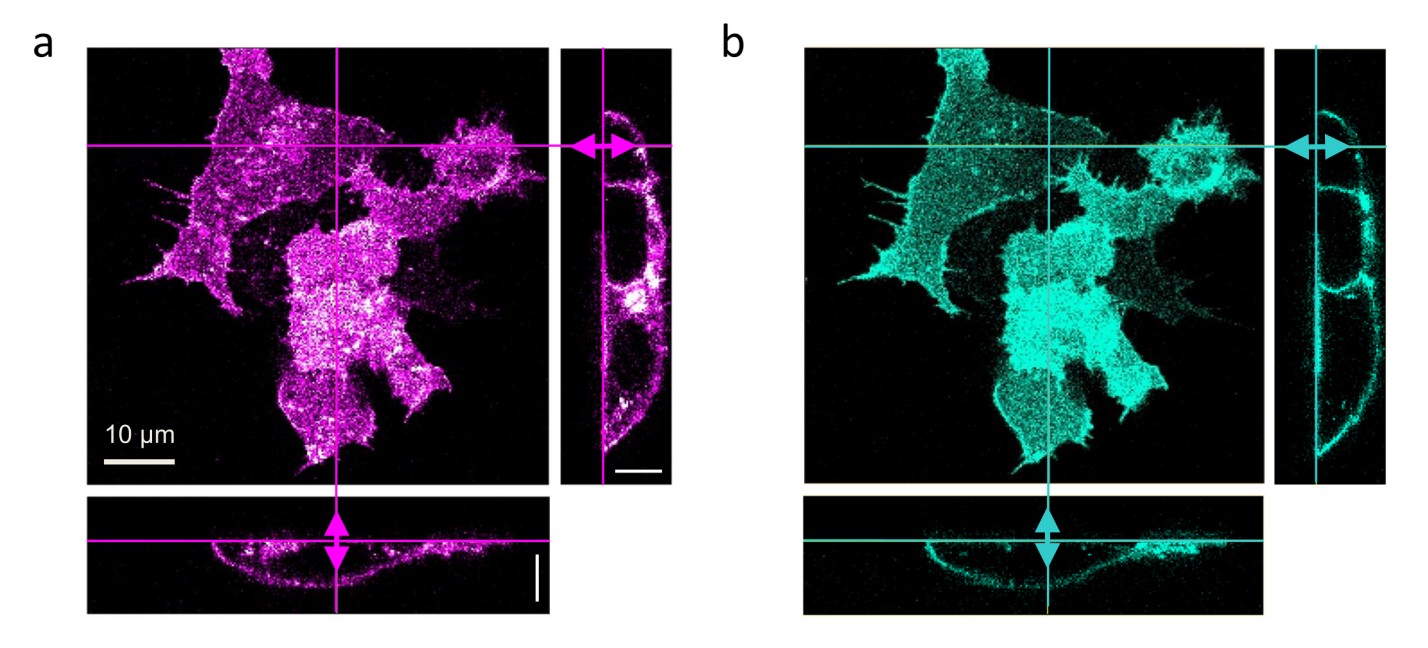

**Figure 3.** Dual-color confocal image stacks of live HEK293T cells, transiently transfected to express LRP6-mCherry (red-fluorescent) and incubated with CM containing 1 nM DKK1-eGFP (green-fluorescent) for 30 min. Confocal images of cells were selected from image stacks taken sequentially (to avoid crosstalk) with 561 (3 µW) and 470 nm (5 µW) laser excitation to visualize (**a**) LRP6-mCherry and (**b**) DKK1-eGFP. A volume of $65 \times 65 \times 25$ µm$^3$ was scanned with $256 \times 256 \times 64$ pixels (pixel dwell time, 40 µs). Shown are a lateral slice near the glass coverslip and cross-sections along the lines marked in the lateral view. Scale bars, 10 µm.

The online version of this article includes the following figure supplement(s) for figure 3:

**Figure supplement 1.** Kinetics of receptor-ligand association and dissociation measured on live cells kept at 37°C and 5% $CO_2$.

**Figure supplement 2.** Expression and activity analysis of endogenously-tagged LRP6-mCherry in CRISPR/Cas9 edited NCI-H1703 cells and exogenous LRP6-mCherry in NCI-H1703 cells with stable gene insertion.

**Figure supplement 3.** Comparison of endogenously tagged LRP6-mCherry and LRP6-tdTomato expressed in CRISPR/Cas9 edited NCI-H1703 cells.

**Figure supplement 4.** Activity tests and western blot analysis of xKremen2-mCherry and xKremen2-eGFP.

**Figure supplement 5.** CRISPR/Cas9 editing of LRP6.

**Figure supplement 6.** Annealed DNA oligonucleotides.

**Figure supplement 7.** Characterization of CRISPR/Cas9 edited NCI-H1703 cell lines harboring endogenous, fluorescently tagged LRP6.

**Figure supplement 8.** siRNA-mediated knockdown to characterize CRISPR/Cas9 edited NCI-H1703 cells.

**Figure supplement 9.** Activity tests and western blot analysis of DKK1-eGFP and DKK2-eGFP.

background fluorescence of live samples and photobleaching of the markers during the experiment can severely compromise the analysis of FCS data. To cope with these adverse effects, we apply appropriate corrections to the data, which we illustrate with an axial lsFCS experiment on a stably transfected cell expressing LRP6-mCherry, exposed to conditioned medium (CM) containing DKK1-eGFP. The axial lsFCS data are visualized as kymographs, 2D plots of the intensity along the scan line versus time (*Figure 4a*). From these data, we determine the membrane positions, $z_{MEM1}(t)$ and $z_{MEM2}(t)$ based on the pixels with the highest photon counts (*Figure 4a*, thick white lines) and compute intensity time traces, $I_{MEM1}(t)$ and $I_{MEM2}(t)$, by integrating each scan over the pixel with the highest photon count and the neighboring two pixels on either side. Finally, the two membrane signals are joined into one raw intensity time trace for each color channel, $I_{raw}(t)$. For background determination, we add the intensities of four regions of one pixel each, ~2 µm (10 pixels) to the left and right of $z_{MEM1}(t)$ and $z_{MEM2}(t)$ (*Figure 4a*, thin white lines). After scaling the background trace to the respective signal trace according to the pixel numbers, we subtract the background from the raw signal trace, $I'(t) = I_{raw}(t) - I_{bg}(t)$. In *Figure 4b*, we have plotted the raw signal intensities, $I_{G,raw}(t)$ and $I_{R,raw}(t)$, and the scaled background intensities, $I_{G,bg}(t)$ and $I_{R,bg}(t)$, in the green and red channels, respectively, during a typical 60-s experiment.

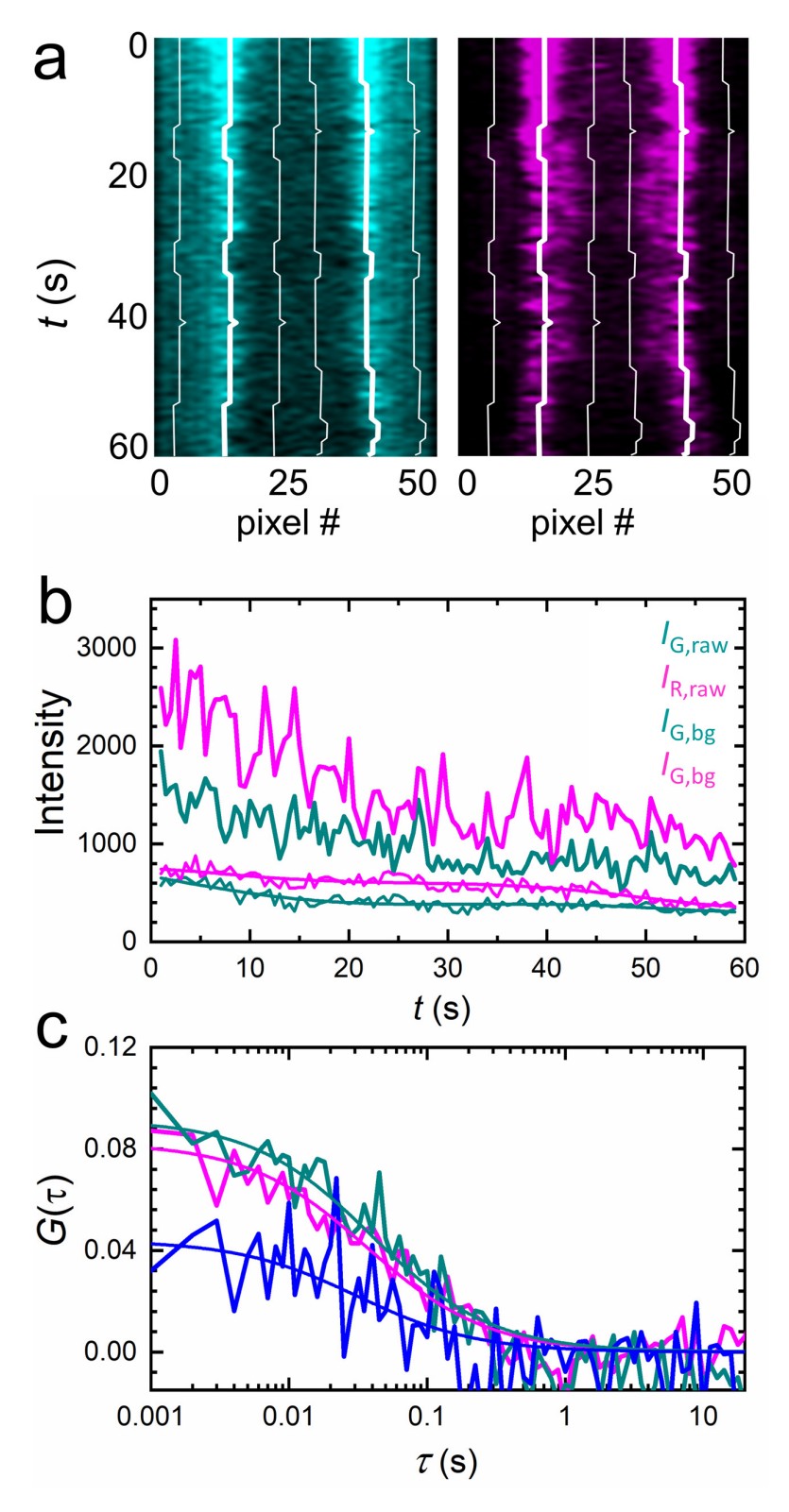

**Figure 4.** Extraction of correlation functions from axial lsFCS intensity time traces on live NCI-H1703 cells stably transfected with LRP6-mCherry and immersed in CM containing 5 nM DKK1-eGFP. (a) Kymograph of the membrane scan recorded in the green (ligand, left) and red (receptor, right) color channels, showing the fluorescence from the membrane during 60 s of axial scanning (500 ms binning time). Thick white lines trace the membrane positions. (b) Time traces of the raw ligand ($I_{G,raw}$) and receptor ($I_{R,raw}$) intensities from the membrane and the corresponding background, $I_{G,bg}$ and $I_{R,}$

*Figure 4 continued on next page*

*Figure 4 continued*

$_{bg}$, determined ±10 pixels away from the membrane (thin white lines in panel a). Lines through, $I_{G,bg}$ and $I_{R,bg}$ are sums of six sine functions fitted to the data for noise filtering. (c) Autocorrelation (cyan, DKK1-eGFP; magenta, LRP6-mCherry) and cross-correlation (blue) functions after background and photobleaching corrections.

The online version of this article includes the following figure supplement(s) for figure 4:

**Figure supplement 1.** Examples of autocorrelation (cyan, DKK1-eGFP; magenta, LRP6-mCherry) and cross-correlation (blue) functions after background and photobleaching corrections, calculated from individual 60 s axial lsFCS intensity time traces.

**Figure supplement 2.** Control experiments demonstrating proper functioning of the axial lsFCS method and analysis, using HEK293T cells transiently transfected to express fluorescently labeled moieties and dual-color fluorescence excitation (PIE) at 470 and 561 nm.

**Figure supplement 3.** Axial lsFCS - data analysis.

A significant loss of signal is apparent in the intensity time traces, indicating that photobleaching continually removes fluorescent markers from the observation area. We compensate the intensity decrease by calculating $I'(0)/I'(t)$ and fitting this ratio with a smooth function, $\gamma(t)$, which we heuristically model as a sum of six sine functions (*Dörlich et al., 2015*). Then, we multiply the intensity time trace by this function, $I(t) = I'(t) \cdot \gamma(t)$. Finally, we calculate intensity autocorrelation functions of the green and red color channels, $G_C(\tau)$ (color index $C = G, R$, respectively), and the cross-correlation function, $G_\times(\tau)$

$$G_C(\tau) = \frac{1}{\langle \gamma(t) \rangle} \cdot \left( \frac{\langle I_C(t) \cdot I_C(t+\tau) \rangle}{\langle I_C(t) \rangle^2} - 1 \right), \tag{3}$$

$$G_\times(\tau) = \frac{\langle I_G(t) \cdot I_R(t+\tau) \rangle}{\langle I_G(t) \rangle \langle I_R(t) \rangle} - 1. \tag{4}$$

The angular brackets denote time averages over the total duration of the experiment. Importantly, the amplitudes of the autocorrelation functions have to be rescaled by $\langle \gamma(t) \rangle^{-1}$ to correct for the loss of fluorophores due to photobleaching. $G_\times(\tau)$, however, does not require rescaling because photobleaching decreases the numerator and denominator in *Equation 4* to equal extents so that the effect cancels. The correlation functions resulting from the single 60-s axial scanning data in *Figure 4a, b* are plotted in *Figure 4c* together with fits using the diffusional model correlation function, *Equation 1*. For illustration purposes, more correlation functions are presented in *Figure 4— figure supplement 1*. These data were measured under systematic variation of receptor density and ligand concentration. We have also included correlation curves of three sets of control experiments on live HEK293T cells as *Figure 4—figure supplement 2*. As a negative control, we have co-expressed our LRP6-mCherry receptor with the membrane marker Mem-eGFP. Both molecules diffuse within the plasma membrane but do not interact. Accordingly, the resulting cross-correlation has zero amplitude (*Figure 4—figure supplement 2a*). As a positive control, we have chosen LRP6-mCherry-eGFP, so that each receptor carries a green and a red fluorescent (or non-fluorescent) protein (*Figure 4—figure supplement 2b*). As another positive control, we have used LRP6-tdTomato, which can be excited at 470 and 561 nm. This experiment is expected to yield complete cross-correlation of the signals from the two color channels, so that the correlation amplitudes differ only due to the wavelength-dependent observation areas (*Figure 4—figure supplement 2c*). Experimental details have been included in the figure caption and Materials and methods section.

To achieve a high precision in $K_D$ determination, the ligand concentration in the CM has to be carefully measured, for example by solution FCS or spectrometrically (for details, see Materials and methods). We dilute the stock solution with cell culture medium, incubate the cells and perform axial lsFCS experiments for several logarithmically spaced ligand concentrations covering the region around $K_D$. This range can easily be identified in confocal images of cultured cells by determining the ligand concentration above which membrane staining by ligands is clearly visible. If $K_D$ values are in the nanomolar range and below, even lower ligand concentrations are required in the CM. Consequently, the kinetics of ligand-receptor formation are slow and sufficient time must be allowed to ensure equilibration before the measurement (*Figure 3—figure supplement 1*).

Fits of the experimental correlation functions $G_G(\tau)$, $G_R(\tau)$ and $G_\times(\tau)$ with **Equation 1** yield the amplitudes, $G_G(0)$, $G_R(0)$ and $G_\times(0)$, for each ligand concentration. We then compute the expression,

$$\Gamma(C_L) = \frac{G_\times(0)^2}{G_G(0)G_R(0)}(C_L) = A\beta \cdot \frac{1}{1 + \frac{K_D}{C_L}}. \tag{5}$$

$\Gamma(C_L)$ is the key function in our quantitative analysis and represents simply a scaled equilibrium binding isotherm, $F(C_L) = (1 + K_D/C_L)^{-1}$. $F(C_L)$ represents the fraction of ligand-bound receptors, which depends on the ratio between the $K_D$ of the ligand-receptor pair and the ligand concentration, $C_L$. The prefactor $A$ depends on the detection areas and is a few percent shy of one for our excitation wavelengths (470 and 561 nm); the ratio between receptors carrying a functional fluorescent domain and all (exogenous or endogenous) receptors capable of specific binding is given by the parameter $\beta$. We note that, in **Equation 5**, we make the (optimistic) assumption that all receptors fused to fluorescent markers are expressed in their functional form, that is that they are competent of binding to their cognate ligand. Likewise, all ligand proteins dispersed in the CM are assumed to be able to recognize the receptor. A derivation and detailed discussion of the validity, assumptions and limitations of **Equation 5** are given in Materials and methods. The parameters $\beta$ and $K_D$ are determined by non-linear least-squares fits of **Equation 5** to the data. Exemplary data on DKK1-eGFP binding to LRP6-mCherry measured on stably transfected NCI-H1703 cells are shown in **Figure 5**. In addition to data and fits of $\Gamma(C_L)$ (**Figure 5a**), we have plotted $F(C_L)$ together with the scaled experimental data (**Figure 5c**). **Figure 5e** displays, for all individual 60-s experiments, the photobleaching parameters $\langle\gamma(t)\rangle$ used to correct the autocorrelation functions (**Equation 3**). In our analyses, we have observed that the function $\Gamma(C_L)$ is very robust against background due to cancellation effects (see Materials and methods). To illustrate this point, we have plotted results for the same experimental data as in **Figure 5a** but calculated without prior background correction of the intensity time traces (**Figure 5b, d, f**). Indeed, there is only a small and insignificant change of the resulting $K_D$ parameter from 0.08 ± 0.01 nM to 0.10 ± 0.02 nM, with and without background correction, respectively. The entire sequence of data analysis steps including data quality management is summarized in **Figure 4—figure supplement 3**.

## DKK1 and DKK2 binding to LRP6-mCherry

To examine the reliability of our axial lsFCS-based analysis for the determination of $K_D$ values of ligand-receptor interactions, we studied DKK1-eGFP binding to NCI-H1703 lung cancer cells expressing LRP6-mCherry in widely differing amounts. To this end, we transiently transfected cells with LRP6-mCherry. Moreover, we used a cell line harboring stably integrated LRP6-mCherry and a CRISPR/Cas9 edited cell line expressing endogenously tagged LRP6-mCherry. A detailed characterization of both cell lines is provided in **Figure 3—figure supplement 2**. HEK293T cells transiently expressing LRP6-mCherry were also included in this comparison. We calculated receptor densities from $G_R(0)$ and the size of the observation area (**Figure 6a**). Note that these figures include only receptors carrying a functional (i.e. fluorescent) mCherry tag; the real numbers are roughly a factor of two greater, assuming a chromophore maturation yield of around 0.5 for mCherry (**Maeder et al., 2007**; **Yasuda et al., 2006**). In any case, it is evident that the different types of gene insertion yield widely different LRP6-mCherry receptor densities at the cell membrane, ranging from 18 μm$^{-2}$ for the CRISPR/Cas9 edited cells to ~1000 μm$^{-2}$ for strongly overexpressing HEK293T cells. As one might expect, the relative scatter of receptor densities is greater for transiently transfected cells than for CRISPR/Cas9 edited and stably transfected cells. The ligand concentration dependencies of the fractional occupancies, $F(C_L)$, are plotted in **Figure 6b**; the corresponding $\Gamma(C_L)$ data (**Equation 5**) are included as **Figure 6—figure supplement 1**. These data reveal a systematic effect of the expression level on the affinity. We find $K_D$ values of 0.08 ± 0.01 and 0.09 ± 0.01 nM for cell lines harboring stably integrated LRP6-mCherry and CRISPR/Cas9 edited cells, whereas three- to fourfold lower affinities are observed for transiently transfected cells. $K_D$ values and other parameters from the analysis are compiled in **Table 1**. For NCI-H1703 cells, the fraction of fluorescent receptors, $\beta$, varies widely, from 14% in the CRISPR/Cas9 cells to 47% in the overexpressing cells. On the one hand, $\beta$ depends on the fraction of endogenously produced (unlabeled) receptors, which are present also in the (haploid) gene edited cells; on the other hand, $\beta$ depends on the ability of the gene

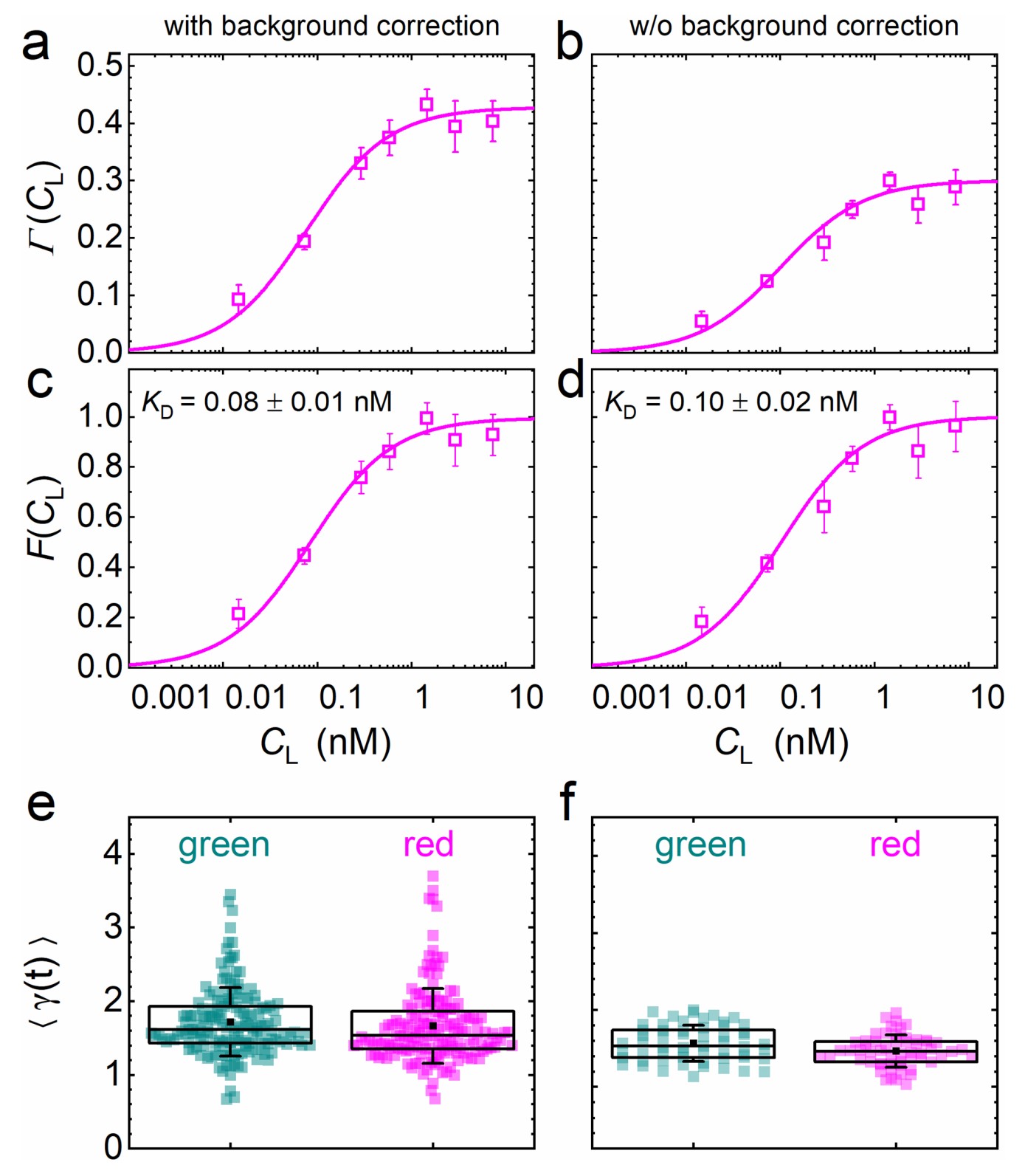

**Figure 5.** Analysis of the concentration dependence of ligand-receptor binding using axial lsFCS on NCI-H1703 cells stably transfected with LRP6-mCherry and immersed in CM containing DKK1-eGFP in varying concentrations. Panels on the left and right show the data with and without background correction, respectively. (a, b) Ligand concentration dependence of the function $\Gamma(C_L)$ defined in *Equation 5*. (c, d) Fractional occupancies, $F(C_L)$, of receptors with DKK1 ligands. Symbols: experimental data (mean ± SEM) from five axial scans of 60 s each; lines: fits with

*Figure 5 continued on next page*

Figure 5 continued

**Equation 5**, yielding the parameters $K_D$ = 0.08 ± 0.01 nM and $\beta$ = 0.44 ± 0.02 with background correction and $K_D$ = 0.10 ± 0.02 nM and $\beta$ = 0.31 ± 0.01 without. (e, f) Parameter $\langle\gamma(t)\rangle$ used for photobleaching correction of the autocorrelation functions. Each point represents an individual 60 s scan. Boxes mark the 25–75% range. The median is indicated by the central line, the mean as a black square; whiskers indicate the SD.

The online version of this article includes the following source data for figure 5:

**Source data 1.** Numerical data of the plots in **Figure 5**.

construct and the cell line to produce fully mature, fluorescent fusion proteins. For strongly overexpressing cells, this latter effect is predominant because their endogenous receptor production can be neglected. Accordingly, the $\beta$ parameter is highest for the transiently overexpressing NCI-H1703 cells. However, a word of caution is in order here. As we discuss in Materials and methods, the amplitude of $\Gamma(C_L)$, which is controlled by $\beta$ in our model, **Equation 5**, in fact also decreases with the maturation yield of the ligand marker and in the presence of non-functional receptors carrying a fluorescent label (see Materials and methods). We believe that this latter issue explains the relatively small amplitudes found for HEK293T cells. The diffusion coefficients ($D_G$, $D_R$) calculated from the correlation decay times of $G_G(\tau)$ and $G_R(\tau)$ (**Table 1**) are around 0.3 – 0.4 µm$^2$ s$^{-1}$. However, their uncertainties are comparatively large, perhaps due to the intrinsic structural heterogeneity of the plasma membrane.

In addition, we have examined possible effects of the bulky fluorescent moiety (eGFP) fused to the DKK1 ligand on the strength of the receptor-ligand interaction. To this end, we have attached an additional SNAP-tag domain (19.4 kDa) to the C-terminus of eGFP, which further increases the bulk of the construct but does not change its fluorescence properties. Axial lsFCS experiments were carried out using HEK293T cells transiently overexpressing LRP6-mCherry. Binding of DKK1-eGFP and DKK1-eGFP-SNAP to the receptors yielded essentially identical results (**Figure 6** and **Table 1**), with $K_D$ = 0.49 ± 0.09 and 0.46 ± 0.09 nM, respectively, suggesting that the fusion tag perturbs the interaction only in a very minor way.

We have further measured the binding of DKK2-eGFP to LRP6-mCherry in the plasma membrane of stably transfected NCI-H1703 cells (**Figure 6—figure supplement 2** and **Table 2**). DKK2 has been reported to have a lower affinity toward LRP6 than DKK1 (**Mao et al., 2001**). Indeed, our result, $K_D$ = 0.26 ± 0.04 nM, for DKK2 indicates a significantly reduced affinity with respect to DKK1 binding to stably transfected NCI-H1703 cells ($K_D$ = 0.08 ± 0.01 nM, **Table 1**).

## DKK1 binding to LRP6-tdTomato

To test the robustness of our analysis, we have further studied the binding of DKK1 to a fusion protein of LRP6 with a distinctly different fluorescent protein marker, tdTomato, emitting at 581 nm. Unlike monomeric mCherry, tdTomato is a tandem dimeric marker protein and thus twice as large as mCherry. It is 5- to 6-fold brighter than mCherry and, therefore, promises a significantly higher SBR in the red channel. We transiently transfected cells with LRP6-tdTomato and we also developed a CRISPR/Cas9 edited cell line expressing endogenously tagged LRP6-tdTomato. A comparison of endogenously tagged LRP6-mCherry and LRP6-tdTomato expressed in CRISPR/Cas9 edited cells is given in **Figure 3—figure supplement 3**. As expected, the calculated density of LRP6-tdTomato in CRISPR/Cas9 edited NCI-H1703 cells was much lower than in transiently transfected cells, with medians at 21 ± 15 and 106 ± 52 µm$^{-2}$, respectively (**Figure 6—figure supplement 3a**). The ligand concentration dependencies of the $\Gamma(C_L)$ data (**Equation 5**) and the corresponding fractional occupancies, $F(C_L)$, are included as **Figure 6—figure supplement 3b, c**; fit parameters are compiled in **Table 2**. For LRP6-tdTomato fusion constructs, we obtained $K_D$ values of 0.10 ± 0.02 and 0.22 ± 0.03 nM for DKK1 binding to CRISPR/Cas9 gene edited and transiently transfected cells, respectively. These values are identical within the error to those obtained for LRP6-mCherry. We also note that, as in the experiments with the mCherry label, the $\beta$ parameter is lower for CRISPR/Cas9 edited than for transiently transfected NCI-H1703 cells, in line with the expectation that the fraction of non-fluorescent receptors in the plasma membranes of CRISPR/Cas9 edited cells is larger. Taken together, our results suggest that (i) the larger size of the tdTomato label does not adversely affect DKK1 binding and (ii) we can reliably measure affinities in the subnanomolar range in live cells even with less bright fluorescent protein tags such as mCherry.

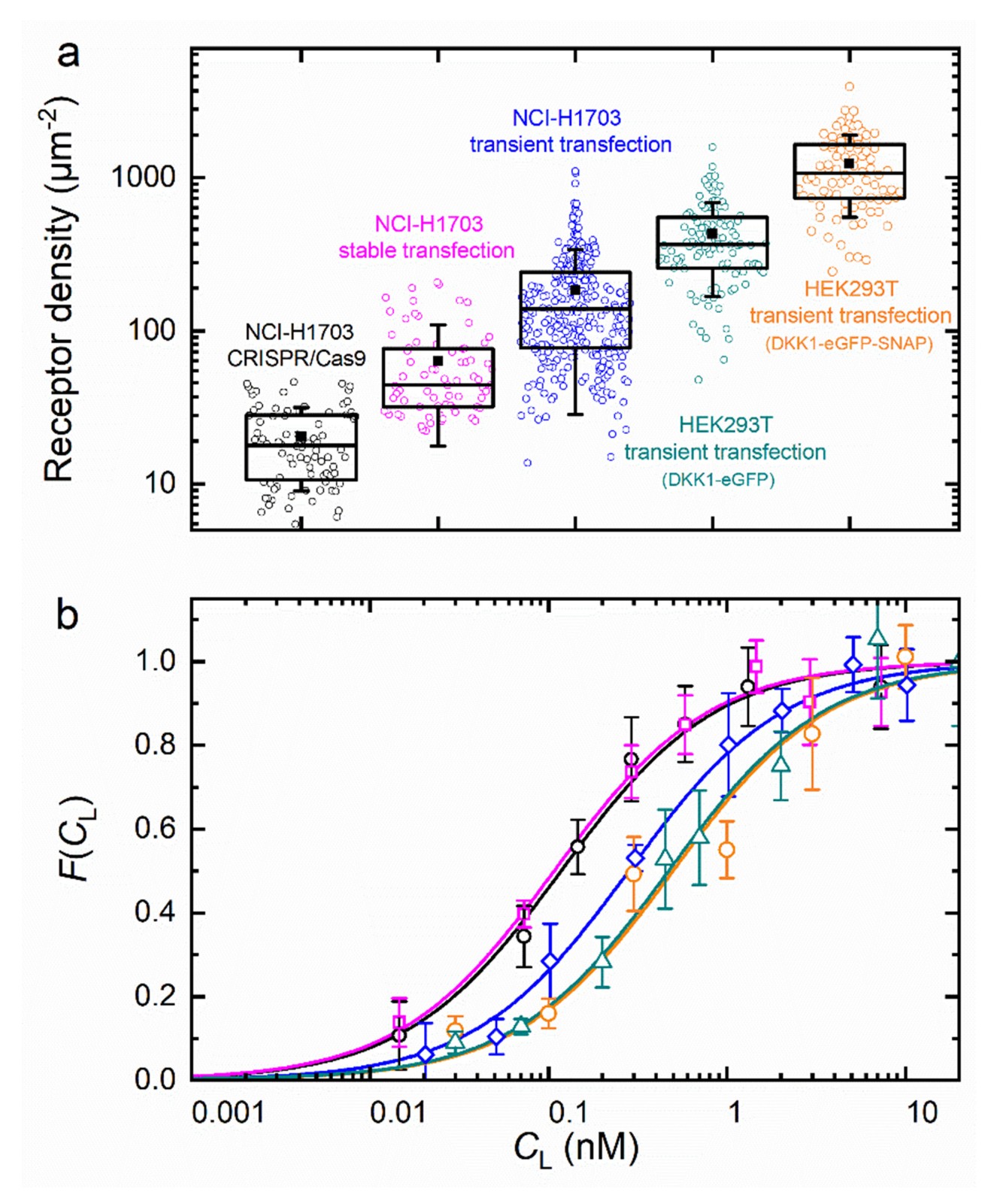

**Figure 6.** Axial lsFCS on transiently and stably transfected as well as CRISPR/Cas9 gene edited NCI-H1703 cells and transiently transfected HEK293T cells, all expressing LRP6-mCherry on their plasma membranes. (a) Densities of fluorescently labeled receptors, determined from the autocorrelation amplitudes, $G_R(0)$, using $\omega_0 = 0.25\ \mu m$. Each data point corresponds to an individual 60-s experiment. Boxes mark the 25 – 75% range; the median is shown as the central line and the mean as a black square. Whiskers indicate the SD. (b) Fractional occupancies, $F(C_L)$, of receptors with Dkk1-eGFP as a
*Figure 6 continued on next page*

*Figure 6 continued*

function of the ligand concentration in the CM. Symbols (shapes/colors as in panel a), experimental data (mean ± SEM from at least three axial scans of 60 s each); lines, fits with *Equation 5*.

The online version of this article includes the following source data and figure supplement(s) for figure 6:

**Source data 1.** Numerical data of the plots in *Figure 6*.

**Figure supplement 1.** Axial lsFCS on HEK293T and NCI-H1703 cells transfected with LRP6-mCherry, measured at 37°C and 5% $CO_2$.

**Figure supplement 2.** Comparison of DKK1-eGFP and DKK2-eGFP binding to NCI-H1703 cells stably transfected with LRP6-mCherry.

**Figure supplement 3.** Axial lsFCS on transiently transfected and CRISPR/Cas9 endogenously tagged NCI-H1703 cells expressing LRP6-tdTomato.

## Interactions between xKremen2-mCherry, DKK1 and LRP6

Kremen proteins (Kremen1 and 2) are type I transmembrane receptors known to modulate Wnt signaling in two distinctly different ways. They were originally identified as negative regulators of Wnt signaling (*Mao et al., 2002*; *Ellwanger et al., 2008*), amplifying DKK-mediated inhibition of Wnt signaling. Formation of a ternary complex of DKK, LRP6 and Kremen was proposed, which is rapidly endocytosed and thereby depletes the Wnt co-receptor LRP6 from the plasma membrane (*Mao et al., 2002*). In the absence of DKK, however, Kremen proteins have also been reported to enhance Wnt signaling by stabilizing the LRP6 receptor at the plasma membrane via Kremen-LRP6 heterodimer formation (*Hassler et al., 2007*). We generated (*Xenopus laevis*) xKremen2-mCherry and xKremen2-eGFP fusion proteins to study Kremen interactions using axial lsFCS. Activity tests confirmed that the presence of mCherry or eGFP appended to the intracellular domain of xKremen2 does not alter its ability to interact with DKK1-eGFP to inhibit Wnt signaling (*Mao et al., 2002*; *Figure 3—figure supplement 4*).

We measured the strength of the Kremen-DKK1 interaction by incubating NCI-H1703 cells transiently expressing xKremen2-mCherry with CM containing DKK1-eGFP in varying concentrations (*Figure 7*). A series of dual-color confocal images showed weak staining of the plasma membrane by DKK1-eGFP binding from the CM at 4 nM but bright staining at 15 nM (*Figure 7—figure supplement 1a*). Consistent with this observation, axial lsFCS yielded a $K_D$ value of 10.3 ± 2.1 nM (*Figure 7*, *Figure 7—figure supplement 2*, *Table 2*), which is considerably higher than the previously reported in vitro values of <3 nM for DKK1 binding to mKremen2 (*Mao et al., 2002*). However, when unlabeled LRP6 was co-expressed together with xKremen2-mCherry, there was pronounced green

**Table 1.** Parameters obtained from the analysis of axial lsFCS data of DKK1 binding to LRP6-mCherry.

| Cell line | HEK293T | | NCI-H1703 | | |
|---|---|---|---|---|---|
| Receptor gene insertion | Transient transfection | | Transient transfection | Stable transfection | CRISPR/Cas9 genome editing |
| Ligand | DKK1-eGFP | DKK1-eGFP-SNAP | DKK1-eGFP | | |
| Receptor density ($\mu m^{-2}$) [*] | 370 ± 150 | 1080 ± 470 | 140 ± 80 | 45 ± 23 | 18 ± 9 |
| $K_D$ (nM) | 0.49 ± 0.09 | 0.46 ± 0.09 | 0.22 ± 0.02 | 0.08 ± 0.01 | 0.09 ± 0.01 |
| $\beta$ | 0.13 ± 0.01 | 0.15 ± 0.01 | 0.47 ± 0.01 | 0.44 ± 0.02 | 0.14 ± 0.01 |
| $D_G$ ($\mu m^2\ s^{-1}$) [†] | 0.44 ± 0.08 | 0.40 ± 0.07 | 0.45 ± 0.11 | 0.29 ± 0.06 | 0.33 ± 0.15 |
| $D_R$ ($\mu m^2\ s^{-1}$) [†] | 0.46 ± 0.09 | 0.45 ± 0.11 | 0.42 ± 0.13 | 0.27 ± 0.05 | 0.33 ± 0.06 |
| $\langle\gamma_G\rangle$ [‡] | 1.32 ± 0.11 | 1.25 ± 0.24 | 1.36 ± 0.21 | 1.62 ± 0.25 | 1.54 ± 0.23 |
| $\langle\gamma_R\rangle$ [‡] | 1.21 ± 0.08 | 1.26 ± 0.13 | 1.33 ± 0.16 | 1.54 ± 0.25 | 1.42 ± 0.29 |

[*] Receptor densities are given as median ± half the range covered by the second and third quartile of the distribution. Individual data points were calculated from receptor autocorrelation amplitudes, $G_R(0)$, and the observation area (0.20 $\mu m^2$).

[†] Diffusion coefficients of receptor-bound ligands and receptors were calculated from the translational diffusion times by using $D = \omega_0^2/4\tau$.

[‡] The photobleaching parameters $\langle\gamma\rangle$ are given as the median value ± half the range covered by the second and third quartile of the distribution compiled from all scans.

The online version of this article includes the following source data for Table 1:

**Source data 1.** Source Data to *Table 1*.

**Table 2.** Parameters obtained from the analysis of axial lsFCS data on NCI-1703 cells.

| Receptor | LRP6-mCherry | LRP6-tdTomato | LRP6-tdTomato | xKremen2-mCherry | xKremen2-mCherry, LRP6 |
|---|---|---|---|---|---|
| Receptor gene insertion | Stable transfection | Transient transfection | CRISPR/Cas9 genome editing | Transient transfection | Transient transfection |
| Ligand | DKK2-eGFP | DKK1-eGFP | DKK1-eGFP | DKK1-eGFP | DKK1-eGFP |
| Receptor density ($\mu m^{-2}$) [*] | 32 ± 16 | 106 ± 52 | 21 ± 15 | 44 ± 33 | 41 ± 30 |
| $K_D$ (nM) | 0.26 ± 0.04 | 0.22 ± 0.03 | 0.10 ± 0.02 | 10.3 ± 2.1 | 0.54 ± 0.06 |
| $\beta$ | 0.43 ± 0.02 | 0.51 ± 0.02 | 0.26 ± 0.01 | 0.45 ± 0.02 | 0.66 ± 0.03 |
| $D_G$ ($\mu m^2\ s^{-1}$) [†] | 0.27 ± 0.08 | 0.35 ± 0.19 | 0.27 ± 0.19 | 0.25 ± 0.15 | 0.32 ± 0.18 |
| $D_R$ ($\mu m^2\ s^{-1}$) [†] | 0.30 ± 0.07 | 0.44 ± 0.34 | 0.39 ± 0.19 | 0.19 ± 0.08 | 0.27 ± 0.09 |
| $\langle \gamma_G \rangle$ [‡] | 1.22 ± 0.17 | 1.31 ± 0.15 | 1.47 ± 0.25 | 1.36 ± 0.22 | 1.30 ± 0.22 |
| $\langle \gamma_R \rangle$ [‡] | 1.41 ± 0.19 | 1.41 ± 0.16 | 1.66 ± 0.39 | 1.42 ± 0.29 | 1.35 ± 0.16 |

[*]Receptor densities were calculated from the amplitudes of the receptor autocorrelation functions, $G_R(0)$, and the observation area (0.20 $\mu m^2$).

[†] Diffusion coefficients of receptor-bound ligands and receptors were calculated from the translational diffusion times by using $D = \omega_0^2/4\tau$.

[‡] The photobleaching parameters $\langle \gamma \rangle$ are given as the median of the distribution from all scans; the errors denote half the width of the second and third quartile of the distribution.

The online version of this article includes the following source data for  Table 2:

**Source data 1.** Source Data to *Table 2*.

staining of the plasma membrane by DKK1-eGFP already at ligand concentrations below 1 nM (*Figure 7—figure supplement 1b*). This effect was not a result of a markedly higher xKremen2-mCherry concentration in the membrane because axial lsFCS revealed an xKremen2-mCherry density of 68 ± 10 $\mu m^{-2}$ with and 79 ± 11 $\mu m^{-2}$ without LRP6 co-expression, respectively (*Figure 7a*). Instead, the apparent affinity of DKK1-eGFP toward xKremen2-mCherry was found to increase roughly 20-fold ($K_D$ = 0.54 ± 0.06 nM) in the presence of co-expressed, unlabeled LRP6 (*Figure 7b*). This finding suggests that LRP6, xKremen2 and DKK1 indeed engage in a ternary complex, with DKK1 binding mediated mainly by the receptor with higher affinity, that is LRP6. DKK1 binding within the LRP6-DKK1-xKremen2 complex is not as strong as to LRP6 alone ($K_D$ = 0.22 ± 0.02 nM, see *Table 1*), presumably because the presence of Kremen interferes with the optimal formation of DKK1-LRP6 binding interfaces. We also note that the $\beta$ parameter, which is normally 0.4–0.5 for transiently transfected NCI-H1703 cells (*Tables 1* and *2*), is exceptionally high (0.66) in this experiment. We would actually have expected a smaller $\beta$, considering that we co-express non-labeled LRP, which can also bind DKK1 and thereby increase the amount of molecules contributing to the intensity of the green channel. We believe that the large $\beta$ value indicates a fraction of complexes that are not simple ternary complexes but larger ones containing two or more xKremen2-mCherry molecules.

The engagement of unlabeled (and thus not visible) LRP6 in the ternary complex was only indirectly inferred from the large change of the apparent binding affinity of DKK1 to xKremen2 upon LRP6 coexpression. Thus, we next aimed to directly observe a LRP6-xKremen2 interaction in the plasma membrane, as earlier suggested (*Hassler et al., 2007*). To this end, we performed axial lsFCS on a large number of NCI-H1703 cells stably expressing LRP6-mCherry and, in addition, xKremen2-eGFP using transient transfection. The FCS data showed almost exclusively that LRP6 was the less abundant of the two receptor species, in accord with a report by *Li et al., 2010* that LRP6 expression is markedly suppressed upon co-transfection of the two receptors. We found that 25 ± 14% of LRP6 receptors co-diffused with the more abundant xKremen2 receptor, that is they were part of a dual-labeled receptor heterodimer; 15 ± 6% of the xKremen2 receptors co-diffused with LRP6 (*Figure 7—figure supplement 3*).

## Discussion

We have developed a fluorescence microscopy-based methodology for the precise quantification of the strength of receptor-ligand interactions on live samples and applied it to the study of receptor-

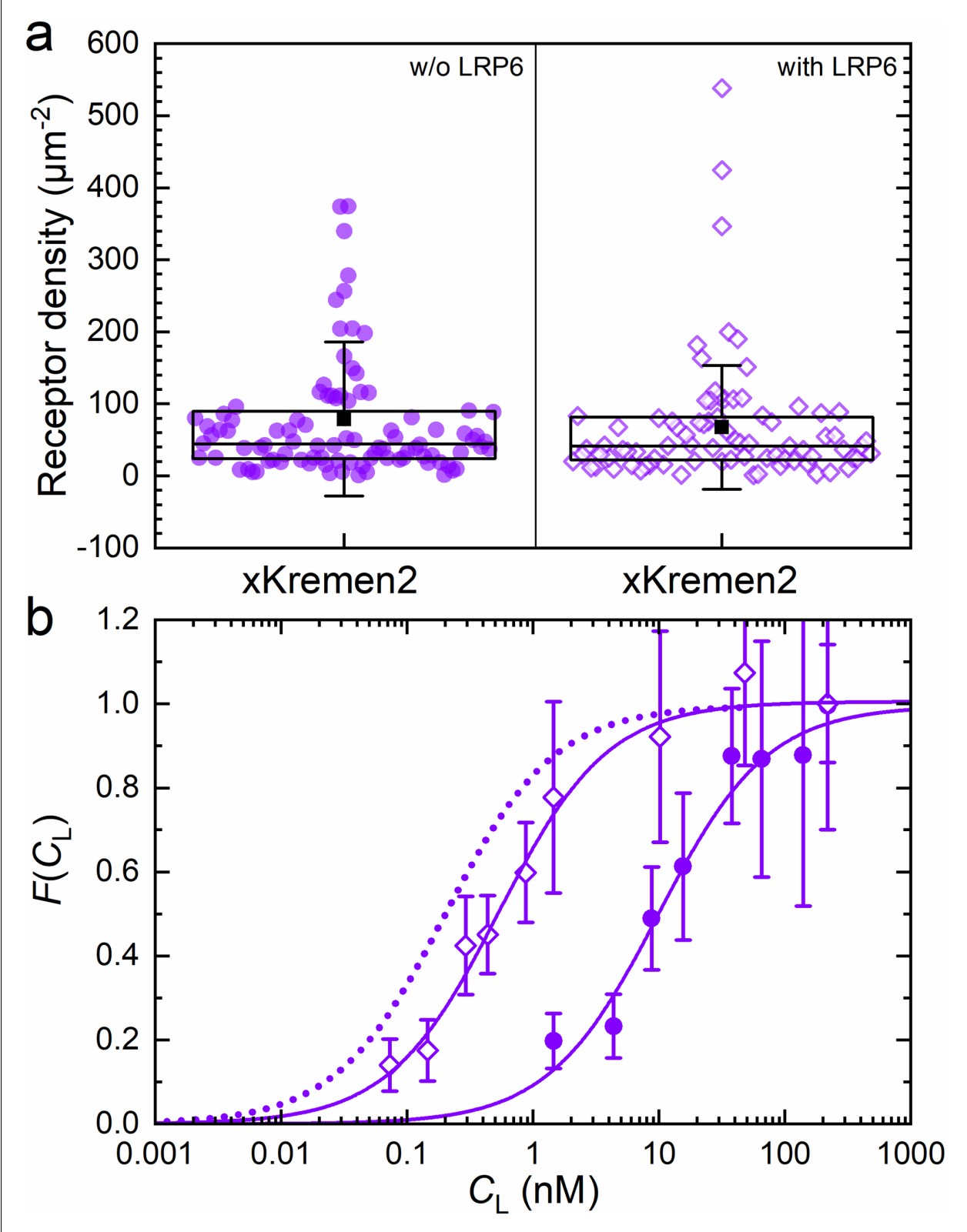

**Figure 7.** Axial line scanning FCS on transiently transfected NCI-H1703 cells expressing xKremen2-mCherry on the plasma membrane. (a) Densities of fluorescently labeled receptors, determined from the autocorrelation amplitudes, $G_R(0)$, with $\omega_0 = 0.25$ µm. Data on the left and right show results without and with LRP6 cotransfection. Each symbol represents data from an individual 60-s experiment. Boxes mark the 25 – 75% range. The median is shown as the central line and the mean as a black square; whiskers indicate the SD. (b) Fractional occupancies, $F(C_L)$, of receptors with DKK1-eGFP as

*Figure 7 continued on next page*

*Figure 7 continued*

a function of the ligand concentration in the CM. Open and closed symbols, experimental data (mean ± SEM from at least three axial scans of 60 s each) from cells with and without LRP6 co-transfection, respectively; solid lines, binding isotherms from fits with *Equation 5*; dotted line, DKK1-eGFP binding curve upon transfection with LRP6 only (for comparison).

The online version of this article includes the following source data and figure supplement(s) for figure 7:

**Source data 1.** Numerical data of the plots in *Figure 7*.
**Figure supplement 1.** Dual-color spinning disk confocal images of live NCI-H1703 cells incubated in CM with four different DKK1-eGFP concentrations (indicated on top).
**Figure supplement 2.** Axial lsFCS on NCI-H1703 cells transfected with xKremen2-mCherry.
**Figure supplement 3.** Apparent fractions of red and green receptors engaged in heterodimeric complexes in the plasma membrane of NCI-H1703 cells stably expressing LRP6-mCherry.

ligand interactions in the Wnt signaling network. In the investigations presented here, we have measured $K_D$ values between 0.08 and 10 nM and we estimate that it will be possible to extend this range by about one more order of magnitude, making the method broadly applicable to a wide range of ligand-receptor interactions. Notably, we are not limited to cultured cells but can also study tissue samples or model organisms.

We have equipped our confocal microscope with a TAG lens that allows ultrafast axial scanning of the observation spot. Here we have exploited this capability for dual-color lsFCS experiments to measure local concentrations of ligand-free and ligand-bound receptors residing in the plasma membrane. Axial scanning offers clear advantages over lateral scanning; most importantly, it features a smaller observation area of circular shape on the plasma membrane of cells. As a consequence, the overall data acquisition time can be reduced by about five-fold with respect to lateral scanning for comparable data precision, and the unavoidable changes in cell position and morphology are less likely to compromise the measurements. For the calculation of autocorrelation and cross-correlation functions, we have implemented a data analysis pipeline that corrects for background and photo-bleaching effects, which is especially important when using fusion constructs with GFP-like proteins for fluorescence labeling. $K_D$ values are determined by fitting the ligand concentration dependence of the ratios of cross- and autocorrelation functions according to *Equation 5*.

Our studies of ligand binding to membrane-bound receptors on living cells have demonstrated that dual-color axial lsFCS in conjunction with our new data analysis is a robust tool to precisely quantify the strengths of ligand-receptor interactions. Data could be collected at receptor densities as low as ca. 20 $\mu m^{-2}$, using, CRISPR/Cas9-edited NCI-H1703 cells expressing endogenously tagged receptors. Wild-type (wt) cells appear to express even lower levels of LRP6 than the endogenously tagged CRISPR/Cas9 edited cells, as judged from comparative western blot analysis. The CRISPR/Cas9 edited cells express the single (heterozygous) targeted allele of LRP6 at higher levels than both LRP6 alleles in wt cells. The endogenously tagged allele of CRISPR/Cas9 edited cells is also expressed at higher levels than the corresponding untagged allele. Possibly, the increased expression of endogenously tagged LRP6 may result from removal and/or displacement of micro-RNA binding sites present in the 3′ untranslated region, which function as negative regulatory elements for LRP6 expression (*Zhang et al., 2018*; *Yang et al., 2019*). In accord with a lower expression from the wild-type LRP6 gene locus, we were unable to detect plasma membrane fluorescence on wt cells upon immersion into DKK-eGFP containing CM at saturating DKK concentrations.

In agreement with earlier in-vitro studies (*Bafico et al., 2001*; *Mao et al., 2001*; *Semënov et al., 2001*; *Bourhis et al., 2010*), we determined subnanomolar equilibrium binding coefficients of eGFP labeled DKK1 and DKK2 ligands binding to LRP6 receptors tagged with mCherry and tdTomato at the C-terminus. The $K_D$ values of DKK1-LRP6 binding were found to increase noticeably with increasing apparent receptor density, from 0.08 nM at 20–50 $\mu m^{-2}$ to 0.5 nM at ~1000 $\mu m^{-2}$. Such an affinity modulation could result from receptor-receptor interactions, which become more prevalent at higher density. One may argue, though, that statistical dispersion of LRP6 receptors in the plasma membrane would lead to an average separation between LRP6 receptors of more than 30 nm at a density of 1000 $\mu m^{-2}$, which appears too large for direct interactions. However, local LRP6 densities can be much higher if the receptors segregate into nanodomains embedded within our observation area of 0.2 $\mu m^2$. Unfortunately, direct visualization of LRP6 clustering is precluded by the diffraction-limited resolution of our confocal microscope, but there is ample evidence for such clustering from a range

of studies. In canonical Wnt signaling, LRP6 not only binds Wnt in a ternary complex with Frizzled receptors but also triggers formation of multi-protein complexes, signalosomes, as part of Wnt pathway activation. Interestingly, a LRP6 variant lacking the entire ectodomain composed of four 300-residue long, concatenated modules (PE1–4), each containing a YWTD β-propeller and an epidermal growth factor (EGF)-like domain, followed by three LDLR type A modules (LA1–3) of 40 residues each, is constitutively active in the absence of Wnt ligands (*Mao et al., 2001*; *Liu et al., 2003*; *Brennan et al., 2004*; *Chen et al., 2014*; *Ren et al., 2015*). Moreover, for a variety of engineered ectodomain variants, the level of signaling activity was found to scale inversely with the size of the ectodomain (*Chen et al., 2014*). *Matoba et al., 2017* argued that there is an intrinsic tendency of the LRP6 transmembrane (TM) domain to oligomerize, leading to spontaneous signaling upon LRP6 overexpression. The curved shape of the bulky ectodomain on the cell surface, which they visualized by negative stain electron microscopy, presumably keeps the TM domains apart and thus counteracts oligomerization and spontaneous signaling. This repulsive effect may be relieved by conformational rearrangements straightening the ectodomain upon formation of LRP6-Wnt-Frizzled complexes (*Matoba et al., 2017*). By contrast, DKK binding would stabilize the bent form of the ectodomain. Thus, in our experiments, a higher LRP6 expression level could presumably lead to denser packing of the bulky LRP6 ectodomain, giving it less freedom to adopt the curved shape favorable for DKK binding, which then may causes the reduced affinity toward DKK antagonists at higher LRP6 expression levels.

We further quantified the interaction of DKK1 with another transmembrane receptor, xKremen2, known to regulate Wnt signaling (*Mao et al., 2002*). With NCI-H1703 cells overexpressing xKremen2, the measured $K_D$ of ~10 nM was much greater than the one previously reported from in-vitro experiments (*Mao et al., 2002*). Notably, upon coexpression of unlabeled LRP6, we observed an affinity increase of roughly 20-fold, implying formation of a complex comprising DKK1, LRP6, and xKremen2, with a much higher DKK1 binding strength. The resultant $K_D$ of 0.54 ± 0.06 nM is closer to but still not equal to the one measured for DKK1-LRP6 binding alone ($K_D$ = 0.22 ± 0.02 nM). These data suggest that the complex is stabilized by DKK1-LRP6 interactions which are reduced if xKremen2 is present, possibly due to steric interference. Notably, strong, synergistic binding of DKK1 to both receptors should lead to a lower $K_D$ than for DKK1-LRP6 binding alone. Thus, the DKK1-LRP6-xKremen complex appears to be mainly stabilized by interactions between DKK1-LRP6 and Kremen-LRP6, but not significantly by the weaker DKK1-xKremen2 interaction ($K_D$ = 10.3 ± 2.1 nM). This scenario implies a mutual interaction between the two receptors, which was confirmed by axial lsFCS experiments on cells expressing both receptors labeled in different colors, showing a high degree of heterodimer formation.

In conclusion, we have introduced a biophysical method based on lsFCS with fast axial scanning. In combination with a sophisticated correlation analysis pipeline, highly precise $K_D$ values can be obtained. Our investigations show that even small changes in $K_D$ values can be measured, attesting to the robustness and reliability of the method. However, besides statistical errors, there are always systematic errors that may deteriorate the accuracy of the results. Two sources of such errors, namely (1) incomplete fluorophore maturation in the fluorescent protein tag, and (2) lack of binding competence of a fraction of ligands or receptors, and their effects on the data have been discussed in the Materials and methods section. To achieve a high accuracy, it is of utmost importance to choose a fluorescent protein tag with optimal maturation yield for the ligands. Moreover, the production of CM should be optimized so as to ensure a large fraction of ligand molecules competent of binding to their cognate receptor. Finally, we note that the accuracy hinges on the precise knowledge of the ligand concentration in the CM. This is not at a trivial issue when dealing with highly dilute solutions in the (sub-)nanomolar range. These and other errors can be minimized by careful experimentation and analysis. Consequently, we feel greatly encouraged to investigate further ligand receptor binding reactions in the Wnt signaling pathway using axial lsFCS.

## Materials and methods

### Microscope setup

Our home-built microscope (see *Figure 1—figure supplement 1* for a schematic) has three picosecond pulsed lasers (L1 – L3) for fluorescence excitation, emitting at 470, 561 nm and 640 nm. They

are externally triggered by 40 MHz transistor–transistor logic (TTL) signals. A delay generator introduces a mutual delay of 12.5 ns between two of the three lasers chosen for dual-channel experiments to enable pulsed interleaved excitation (*Müller et al., 2005*), so that the fluorescence excited by the two lasers can be detected in two separate time windows of 12.5 ns width. The laser beams are coupled into a wideband fiber coupler (WFC) by a mirror (M) and two dichroic mirrors LP1 and LP2. To minimize axial misalignment of the foci, the combined beam is split into three paths via a wavelength division multiplexer (WDM), and further combined using a mirror M and dichroic beamsplitters LP3 and LP4. In each path, the axial focus position of the beam is carefully adjusted by changing the distance between the fiber end and the coupling lens. After reflection by quad-band mirror QB, the excitation light passes through a TAG lens enabling fast axial scanning of the excitation focus, the laser scanner, scan lens (SL), and enters a Leica DMi8 microscope frame (Leica Microsystems, Wetzlar, Germany), in which the beam is collimated by a tube lens (TL) and reflected by a mirror (M) into the vertical direction. A quarter-wave plate generates circular polarization to avoid polarized excitation. The laser beam is finally focused into the sample by a 63×/1.2 NA water immersion objective. For additional, slow scanning in the axial direction, the objective can be moved by a motorized linear actuator with an accuracy of 0.5 nm (M-122, Physik Instrumente, Karlsruhe, Germany) and a piezo actuator (P-720, Physik Instrumente).

The fluorescence emitted by the sample propagates back through QB and is coupled into a multi-mode fiber serving as a pinhole of 1 AU (for 640 nm light). Subsequently, it is separated into two color channels by a 555 nm longpass dichroic mirror LP5, passed through bandpass filters BPF1 and BPF2 and are finally detected by two avalanche photodiodes APD1 and APD2. A TCPSC card (SPC-150, Becker and Hickl GmbH, Berlin, Germany) records, for each photon, its absolute arrival time and its delay time with respect to the 40-MHz laser trigger and, furthermore, the lateral $(x, y)$ positions of the galvanometer scanner as well as the timing signal, $t_{TAG}$, issued by the TAG lens control unit for $z$ coordinate determination.

## Focus size calibration

Multi-color beads (TetraSpeck microspheres, diameter $d$ = 0.1 µm, Thermo Fisher Scientific, Karlsruhe, Germany) were sparsely immobilized on a chambered glass plate and scanned with the focused 470 nm and 561 nm laser beams (both 5 µW). The emitted light was passed through Brightline bandpass filters (HC 525/50 for green emission, HC 600/37 for red emission, Semrock, Rochester, NY) before detection. In each color channel, ten beads were selected for focus size determination. By analyzing the intensity distributions across their centers and correcting the measured full widths at half maximum (FWHM$_m$) values for the actual size of the beads according to FWHM = $(FWHM_m^2 − d^2)^{1/2}$, we obtained FWHM$_{470nm}$ = 242 ± 4 nm (mean ± standard error of the mean (SEM)) and FWHM$_{561nm}$ = 295 ± 9 nm (mean ± SEM). The focus parameter, $\omega_0$, which corresponds to the distance over which the intensity decays by a factor of $1/e^2$, was calculated by using the relation $\omega_0 = (2 \ln(2))^{1/2}$ FWHM, yielding $\omega_0$ = 207 ± 3 nm and 251 ± 8 nm for 470 nm and 561 nm laser excitation (see *Figure 2—figure supplement 1*).

## Dependence of the precision of axial and lateral lsFCS on the duration of the measurement

The overall data acquisition time needed to obtain a FCS curve with sufficient quality is an important parameter, especially when studying live samples which unavoidably move during the experiment. To address this issue, we have performed 14 axial and lateral line scanning FCS experiments on Atto647N-labeled GUVs, each with a duration of 400 s. From the intensity time traces, 40 individual $G_i(\tau)$ curves were calculated, normalized to amplitude 1 and averaged to yield a single $\bar{G}(\tau)$. The scatter of an individual curve around the average was quantified by the standard deviation (SD),

$$SD_i = \sqrt{\frac{1}{\tau} \sum_{\tau} (G_i(\tau) - \bar{G}(\tau))^2}. \tag{6}$$

This calculation was carried out for all 40 curves, and the average and SD were calculated for this ensemble. Subsequently, the procedure was repeated with the same but artificially shortened intensity time traces to generate the data presented in *Figure 2e*. This graph shows that SD ≈ 0.1 can be reached with axial lsFCS in one-fifth of the time needed with lateral lsFCS.

## FCS data acquisition and analysis

Axial lsFCS data were collected on live cells for 60 s with 470 nm excitation of eGFP at 5 µW (2.8 kW/cm$^2$) and 561 nm excitation of mCherry/tdTomato at 3 µW (1.5 kW/cm$^2$). Cells were kept at 37°C and 5% $CO_2$. Before data collection, the cell culture medium was replaced by 500 µl CM with DKK1-eGFP or DKK2-eGFP concentrations, $C_L$, in the range of 0.01 nM – 220 nM. To ensure that binding equilibrium was reached prior to the axial lsFCS experiment, at $C_L$ = 3.5 nM, the cells were incubated for 5 min (see *Figure 3—figure supplement 1*). For all other ligand concentrations, the incubation times were adjusted by the ratio 3.5 nM/$C_L$, as expected for a bimolecular reaction. At concentrations below 1 nM, the CM was replaced by fresh medium in the middle of the incubation period to compensate for ligand depletion from the CM due to receptor binding or endocytosis.

At each $C_L$, six cells were selected on average for axial lsFCS on both the basal and the top membranes (~50% each). With the TAG lens turned off, the focal volume was centered on the membrane under study. Then, the TAG lens was turned on (at 75% of its maximum amplitude) to scan the focus across the membrane, spanning roughly 5.3 µm in axial direction. Photons were registered by the TCSPC unit and sorted according to their macroscopic and microscopic arrival times to generate intensity time traces in the two color channels used for correlation analysis.

These data were plotted as kymographs, that is 2D plots of the intensity along the scan line (on the x-axis) line by line for the entire scan sequence and, thus, as a function of time (on the –y-axis). Data sets showing pronounced bright features near the membrane (fluorescent clusters) or strongly fluctuating membrane positions were discarded. Furthermore, kymographs were inspected for correct focusing, and if the plasma membrane was not near to the middle of the axial oscillation range, data were also discarded. After correcting the remaining intensity time traces for background and photobleaching, autocorrelation and cross-correlation curves were calculated with *Equations 3 and 4*, respectively, and fitted with the diffusional model correlation function, *Equation 1*. Those data sets for which the fit returned markedly deviant correlation times, $\tau_D$, or different ones for the two-color channels were discarded. From the amplitudes, $G_G(0)$, $G_R(0)$ and $G_\times(0)$, the function $\Gamma(C_L)$ was computed (*Equation 5*). At least three independent $\Gamma(C_L)$ values were determined and averaged for each particular ligand concentration, $C_L$, plotted as a function of ligand concentration and fitted with *Equation 5* by variation of two parameters, $K_D$ and $\beta$ (see below). The entire sequence of data analysis steps including data quality management is shown diagrammatically in *Figure 4—figure supplement 3*.

To ensure proper functioning of the axial lsFCS apparatus and the analysis, we frequently performed control experiments with samples designed to give a sizeable (or complete) cross-correlation or none at all. Examples are shown in *Figure 4—figure supplement 2*.

## Determination of $K_D$ from ligand concentration-dependent dual-color axial lsFCS

To measure the equilibrium dissociation coefficient of a ligand-receptor pair, we take axial lsFCS data over a wide range of free ligand concentrations on cells expressing the receptor. In the following, we assume that all receptors on the cell surface are capable of binding their cognate ligand, and that the ligands in turn are all competent of binding to their cognate receptor. We further assume that the ligands are labeled with fluorescent proteins emitting in the green channel and have a fluorophore maturation yield, $\eta = C_L / (C_L + C_l) = 1$, with $C_L, C_l$ denoting ligands carrying a functional (fluorescent) and non-functional (dark) marker protein, respectively. The extension to $\eta < 1$ is straightforward and will be discussed below. By using regular solution FCS, we can precisely determine $C_L$ of the CM stock solution, from which a dilution series with a range of ligand concentrations is generated. Ligands in the medium bind to receptors on the cell membrane to form ligand-receptor complexes, which are present at concentrations (area densities) $C_{RL}$ (receptor fluorescent) and $C_{rL}$ (receptor nonfluorescent). Note that receptors may be nonfluorescent because the cells endogeneously produce (non-fluorescent) receptors, or the fluorescent moeity of the fusion protein did not mature to its functional form. Furthermore, depending on the ratio of $K_D$ and $C_L$, there will be a smaller or larger fraction of fluorescently labeled receptors at a concentration, $C_R$, that do not have a ligand bound.

For the analysis, it is important that fast 3D diffusion of the ligand in the medium and much slower 2D diffusion of the ligand in ligand-receptor complexes in the membrane are well separated

in time. Thus, only slow 2D diffusion with characteristic times of milliseconds needs to be considered. Two autocorrelation functions, $G_G(\tau)$ and $G_R(\tau)$, and one cross-correlation function, $G_\times(\tau)$, can be calculated from our dual-color axial lsFCS data and modeled as

$$G_G(\tau) = \frac{1}{A_G(C_{rL} + C_{RL})^2} \cdot (C_{rL} + C_{RL}) \left(1 + \frac{4D_{RL}\tau}{\omega_G^2}\right)^{-1},$$ (7)

$$G_R(\tau) = \frac{1}{A_R(C_R + C_{RL})^2} \cdot \left(C_R \left(1 + \frac{4D_R\tau}{\omega_R^2}\right)^{-1} + C_{RL} \left(1 + \frac{4D_{RL}\tau}{\omega_R^2}\right)^{-1}\right),$$ (8)

$$G_\times(\tau) = \frac{1}{A_{eff}(C_R + C_{RL}) \cdot (C_{rL} + C_{RL})} \cdot C_{RL} \left(1 + \frac{4D_{RL}\tau}{\omega_{eff}^2}\right)^{-1}.$$ (9)

Here, $A_G = \pi\omega_G^2$ and $A_R = \pi\omega_R^2$ are the confocal observation areas of the green and red color channels, respectively; $A_{eff} = \pi\omega_{eff}^2$ is the effective area in the dual-color experiment, with $\omega_{eff}^2 = (\omega_G^2 + \omega_R^2)/2$. The diffusional correlation times of the different species are introduced as $\tau_D = \omega_{G/R}^2/4D$.

For $K_D$ determination, we are only interested in the concentrations of the different species and, thus, the amplitudes of the correlation functions at $\tau = 0$. We calculate the ratios

$$\frac{G_\times(0)}{G_G(0)} = \frac{A_G}{A_{eff}} \cdot \frac{C_{RL}}{(C_R + C_{RL})},$$ (10)

$$\frac{G_\times(0)}{G_R(0)} = \frac{A_R}{A_{eff}} \cdot \frac{C_{RL}}{(C_{rL} + C_{RL})},$$ (11)

and multiply both equations to obtain

$$\frac{G_\times(0)^2}{G_G(0)G_R(0)} = \frac{A_G A_R}{A_{eff}^2} \cdot \frac{C_{RL}}{(C_R + C_{RL})} \cdot \frac{C_{RL}}{(C_{rL} + C_{RL})}.$$ (12)

In *Equation 12*, the observation areas appear in the form of the squared ratio of the geometric and arithmetic means, $A = (A_G A_R)/A_{eff}^2$, which equals 0.967 for the observation areas of our dual-color FCS experiments with 470 and 561 nm laser excitation. In general, this parameter is just a few percent shy of one for realistic observation areas in a dual-color experiment. Thus, the geometric differences between the two color channels almost cancel if we ensure perfect overlap between the observation areas for the two colors. Notably, in lateral lsFCS and 3D FCS dual-color cross-correlation measurements, optimal overlap of the observation areas and volumes, respectively, is more difficult to achieve because of mutual displacements along the axial direction. This problem is completely absent in axial line-scanning FCS due to its intrinsic axial self-alignment.

We introduce the equilibrium dissociation coefficient, $K_D = C_R C_L/C_{RL}$, and rearrange *Equation 12* to read

$$\Gamma(C_L) = \frac{G_\times(0)^2}{G_G(0)G_R(0)}(C_L) = A\beta \cdot \frac{1}{1 + \frac{K_D}{C_L}}.$$

$\Gamma(C_L)$ takes the form of a binding isotherm, multiplied by a scaling factor which includes, apart from the geometric parameter $A$, the maturation yield of the red fluorophores marking the receptor, $\beta = C_{RL}/(C_{rL} + C_{RL})$. Thus, the amplitude of $\Gamma(C_L)$ for $C_L \rightarrow \infty$ yields an estimate of this value. Notably, incomplete fluorophore maturation of the receptor tag affects only the amplitude of the binding isotherm but not the $K_D$ value.

Up to this point, we have assumed that the maturation yield of the fluorescent protein marking the ligand is one. How does *Equation 5* change upon generalization to $\eta < 1$? In this case, the autocorrelation amplitude of the red channel, $G_R(0)$, is not only governed by ligand-free receptors but also those having a dark ligand bound. Then, $K_D$ in *Equation 5* is not equal to $C_R C_L/C_{RL}$ but rather

$$\frac{(C_R + C_{Rl})C_L}{C_{RL}} = K_D + \frac{C_{Rl}}{C_{RL}}C_L = K_D + \frac{(1-\eta)}{\eta}C_L. \tag{13}$$

Substituting $K_D$ in *Equation 5* by the right-hand side of *Equation 13* yields

$$\Gamma(C_L) = A\beta\eta \cdot \frac{1}{1 + \frac{\eta K_D}{C_L}}. \tag{14}$$

*Equation 14* reveals that incomplete maturation of the ligand marker affects the binding curve in two ways: $\eta$ becomes part of the overall scaling factor of the binding curve (y-axis) and decreases its amplitude. More importantly, in contrast to fluorophore maturation effects of the receptor tags, $\eta$ also affects the location of the binding step along the $C_L$ coordinate (x-axis); the midpoint of the binding isotherm is at $C_L = \eta K_D$ and not at $K_D$. Alternatively, one can view this modification as a redefinition of the x-axis to represent the total ligand concentration, $C_L + C_l = C_L/\eta$. Thus, the presence of dark ligand markers affects the $K_D$ values. We note that maturation yields are difficult to determine, and are by no means characteristic parameters of the chosen fluorescent marker proteins. They generally vary with the conditions of the experiment including the nature of the expression system, expression temperature and yield, and details of the design of the fusion construct carrying the marker. To minimize this source of systematic error, we have chosen eGFP, a protein for which high maturation yields (80 – 100%) have been reported (*Foo et al., 2012b*; *Komatsubara et al., 2019*; *Sniegowski et al., 2005*; *Ulbrich and Isacoff, 2007*).

In addition to incomplete fluorescence marker maturation, there is another source of systematic error relevant to most methods of affinity measurement (with the exception of single molecule-based experiments, for example by *Manz et al., 2017*). A fraction of receptors on the plasma membrane as well as a fraction of ligands in the CM may be incompetent of binding, for example due to improper folding. Unlike the degree of fluorophore maturation, which can at least be estimated, quantification of the functional fraction is a formidable task. How does this problem affect the results of our axial lsFCS experiments? Let us start with the effect of non-binding ligands present in the CM. In this case, their concentration will be less than the one anticipated from the FCS amplitudes or from any spectroscopic method capable of concentration determination. Accordingly, we have to rescale the $C_L$ axis of our binding curves to represent only the functional ligand fraction, which leads to real $K_D$ values that are smaller than the apparent ones, that is when assuming 100% binding-competent ligands. The effect of binding-incompetent receptors being present is a little more involved and we only sketch the consequences here. We have to include an additional species, non-functional receptors, $C_{R\dagger}$, in $G_R(\tau)$ in *Equation 8*. If we then calculate $\Gamma(C_L)$, yet another scaling factor appears in the amplitude of the binding isotherm, which represents the fraction of functional receptors, $\rho = (C_R + C_{RL})/(C_{R\dagger} + C_R + C_{RL})$. Importantly, in analogy to the case of incomplete fluorophore maturation of the receptor tag, the presence of receptors incapable of ligand binding only modifies the amplitude of the binding curve but has no effect on $K_D$.

Throughout this work, we have used *Equation 5* for our data analysis. The parameters $\beta$ and $K_D$ were determined by non-linear least-squares fits of $\Gamma(C_L)$ to the experimental data. If the maturation yield of the ligand marker is known, one may choose *Equation 14* or simply replace $K_D$ obtained from fitting with *Equation 5* with $K_D/\eta$ to correct for incomplete maturation of the ligand tag. Likewise, if the functional fractions are known, one can take them into acount as described above.

Finally, without going into detail, we note that the combination of correlation functions through which $\Gamma(C_L)$ is defined not only leads to rather simple expressions for the analysis of $K_D$ but also has the beneficial property that background contributions are largely cancelled even if explicit background correction of the intensity time traces is omitted (*Figure 5*). We refer interested readers to papers that discuss correction factors for pair correlation function amplitudes due to uncorrelated background (*Weidemann et al., 2002*; *Foo, 2012a*). From the equations presented in these papers, it can be easily verified that the correction factors indeed cancel in the calculation of $\Gamma(C_L)$.

## Analysis of LRP6-xKremen2 heterodimer formation using dual-color axial lsFCS

To estimate the fraction of LRP6 molecules bound to xKremen2, we took axial lsFCS data on stably LRP6-mCherry expressing NCI-H1703 cells that were co-transfected with the xKremen2-eGFP

plasmid. Importantly, background expression of endogenous (unlabeled) Kremen and LRP6 is negligible in NCI-H1703 cells. In the following analysis, we refer to xKremen2-eGFP as receptor $G$ (emissive green form) or $g$ (non-emissive green form) and, accordingly, to LRP6-mCherry as receptor $R$ (emissive red form) or $r$ (non-emissive red form). These species may be present in the plasma membrane either as monomers or as four different heterodimer species, $GR$, $gr$, $gR$ and $Gr$. The amplitudes of the relevant pair correlation functions are given by

$$G_G(0) = \frac{1}{A_G(C_{GR} + C_G + C_{Gr})} = \frac{1}{A_G(\eta_R^{-1} C_{GR} + C_G)}, \tag{15}$$

$$G_R(0) = \frac{1}{A_R(C_{GR} + C_R + C_{gR})} = \frac{1}{A_R(\eta_G^{-1} C_{GR} + C_R)}, \tag{16}$$

$$G_\times(0) = \frac{C_{GR}}{A_{eff}(C_{GR} + C_R + C_{gR}) \cdot (C_{GR} + C_G + C_{Gr})} = \frac{C_{GR}}{A_{eff}(\eta_G^{-1} C_{GR} + C_R) \cdot (\eta_R^{-1} C_{GR} + C_G)}. \tag{17}$$

Here we have introduced the maturation yields, $\eta_G = C_G / (C_G + C_g)$ and $\eta_R = C_R / (C_R + C_r)$, that is the fractions of eGFP and mCherry proteins, respectively, carrying a functional fluorophore. The observation area-weighted ratios of cross- and autocorrelation functions, $F_R'$ and $F_G'$, are given by the following expressions,

$$F_R' = \frac{A_{eff}}{A_G} \cdot \frac{G_\times(0)}{G_G(0)} = \frac{C_{GR}}{\eta_G^{-1} C_{GR} + C_R} \quad \Longrightarrow \quad F_R = \frac{C_{GR}}{C_{GR} + C_R} = \frac{1}{F_R'^{-1} + (1 - \eta_G^{-1})}, \tag{18}$$

$$F_G' = \frac{A_{eff}}{A_R} \cdot \frac{G_\times(0)}{G_R(0)} = \frac{C_{GR}}{\eta_R^{-1} C_{GR} + C_G} \quad \Longrightarrow \quad F_G = \frac{C_{GR}}{C_{GR} + C_G} = \frac{1}{F_G'^{-1} + (1 - \eta_R^{-1})}, \tag{19}$$

with $A_{eff}/A_G$ = 1.23 and $A_{eff}/A_R$ = 0.84 under our experimental conditions. $F_R'$ and $F_G'$ correspond to bound fractions only for 100% fluorophore maturation but, as shown above, allow one to calculate the true bound fractions, $F_R$ and $F_G$, provided that the maturation yields are known. According to *Equations 18 and 19*, the experimentally determined average $F_R' = 0.25$ (*Figure 7—figure supplement 3*) corresponds to a bound fraction of LRP6 receptors, $F_R = 0.26$, assuming $\eta_G = 0.9$ for eGFP. Lower maturation yields result in larger relative corrections. Thus, the experimentally determined average $F_G' = 0.15$ yields $F_G = 0.18$, assuming $\eta_R = 0.5$ for mCherry.

## Giant unilamellar vesicles

GUVs were prepared according to the protocol by *García-Sáez et al., 2010* using a lipid mixture consisting of 50 mol% sphingomyelin (SM, Sigma-Aldrich, Darmstadt, Germany), 49.999 mol% cholesterol (Sigma-Aldrich) and 0.001 mol% 1,2-dipalmitoyl-sn-glycero-3-phosphoethanolamine (DPPE) labeled with Atto 647N (Sigma-Aldrich).

## Plasmids

*pCS2$^+$hLRP6-tdTomato* was generated by replacing the *mCherry* open reading frame (ORF) of *pCS2$^+$hLRP6-mCherry* (*Dörlich et al., 2015*) with the ORF of *tdTomato*, using the XbaI and SnaBI restriction sites. For the cell line stably expressing LRP6-mCherry, the ORF of the *hLRP6-mCherry* fusion was inserted between the NheI/NotI restriction sites of the *pEF1a-IRES-Neo* plasmid (a gift from Thomas Zwaka, Addgene plasmid # 28019) to create *pEF1a-hLRP6-mCherry-IRES-Neo*. *pCS2$^+$V5-xKremen2-mCherry* was generated by replacing the *hLRP6* ORF of *pCS2$^+$hLRP6-mCherry* with the ORF of *xKremen2* using NEBuilderHiFi DNA Assembly (New England Biolabs E2621S). *pCS2$^+$hDKK1-eGFP* and *pCS2$^+$hDKK2-eGFP* were described previously (*Krupnik et al., 1999*; *Brott and Sokol, 2002*; *Dörlich et al., 2015*). *pCS2$^+$hDKK1-eGFP-SNAP* was generated from *pCS2$^+$hDKK1-eGFP* by adding 36 bases (GGTTCAGCAGGTTCTGCAGCAGGTTCTGGAGAGTTC) coding for a 12-amino acid linker (GSAGSAAGSGEF) and the gene of the SNAP-tag domain (a kind gift from Professor Nils Johnsson, University of Ulm, Germany) using the NEBuilderHiFi DNA

Assembly kit (New England Biolabs). The *pCS2+hLRP6-mCherry-eGFP* construct was built in an analogous fashion, using the eGFP instead of the SNAP-tag gene.

## Generation of cell lines stably expressing fluorescently tagged LRP6

NCI-H1703 cells were transfected with 2 μg *pEF1a-hLRP6-mCherry-IRES-Neo* in six-well plates using ScreenFectA (ScreenFect GmbH, Eggenstein, Germany) according to the manufacturer's one-step protocol. 24 hr post-transfection, the medium was exchanged, and 48 hr post-transfection one tenth of the cells were transferred to 10 cm$^2$ dishes and cultured in RPMI 1640 supplemented with 700 μg/ml G418 (Sigma-Aldrich). Selection was performed for 12 d. Clonal lines were obtained by limited dilutions in 96-well plates.

## Generation of CRISPR/Cas9 cell lines expressing fluorescently tagged LRP6

The human LRP6 locus was targeted within exon 23 for C-terminal endogenous tagging with a fluorescent protein (see *Figure 3—figure supplement 5*). Using the E-CRISP online tool (*Heigwer et al., 2014*), we identified a 20 bp protospacer (5'-ACGTTGGAGGCAGTCAGAGG-3') followed on the 3' end by a NGG PAM (protospacer adjacent motif) downstream of the LRP6 stop codon, in exon 23. DNA oligonucleotides were designed and annealed to generate the following short double-stranded DNA fragment containing the 20 bp protospacer, an additional guanine nucleotide at the 5'end that increases targeting efficiency (*Cong et al., 2013*) and 4 bp overhangs suitable for ligation into a BbsI restriction site (*Figure 3—figure supplement 6*). This short double-stranded DNA fragment was ligated into the BbsI restriction site of the pX330-U6-Chimeric_BB-CBh-hSpCas9 (D10) vector (*Cong et al., 2013*), generating a plasmid encoding a single guide RNA (sgRNA) harboring the *LRP6* specific 20 bp protospacer under the control of a U6 RNA Polymerase III promotor.

The PCR2.1-PITX-eGFP-PGK-Puro vector (*Hockemeyer et al., 2011*) was used as a template to create the PCR2.1-hLRP6-mCherry-PGK-Puro and PCR2.1-hLRP6-tdTomato-PGK-Puro donor vectors. DNA sequences of ~700 bp for the left and the right homology arms (LHA, RHA) of LRP6 were PCR-amplified from genomic DNA using the primer pairs:

LHA-F 5'-AAACCTGCAGGCCCATTCTTCCCATGGG-3',
LHA-R 5'-AAAGCTAGCGGAGGAGTCTGTACAGGG-3'; and
RHA-F 5'-AAAGGCGCGCCTGAGGAGGGGCCCTCCTC-3',
RHA-R 5'-AAAGGCCGGCCCAGAGACTTGCACACACACAGC-3'.

The homology arm PCR products were cloned into the PCR2.1 donor vector using the SBfI/NheI (LHA) or AscI/FseI (RHA) restriction sites. The PCR amplified mCherry and tdTomato ORFs were cloned into the PCR2.1 donor vector using the NheI/EcoRI restriction sites. For the LRP6-mCherry donor vector, complementary single strand oligonucleotides bearing a 12-amino acid linker (ASTLEASLERAS) were annealed and cloned into the NheI digested vector between LHA and mCherry. For the LRP6-tdTomato donor, the NheI restriction site (AS) served as a linker between LRP6 and tdTomato. In order to protect the donor vector from cleavage by the Cas9 nuclease, a silent mutation within the PAM site of the right homology arm of LRP6 was introduced using the QuickChange Lightning Site-directed Mutagenesis Kit (Agilent Technologies, Waldbronn, Germany).

To establish a cell line expressing endogenously tagged LRP6, NCI-H1703 cells cultured in 6-well plates were transfected with 1 μg donor plasmid and 1 μg CRIPSPR/Cas9 plasmid using ScreenFectA (ScreenFect GmbH) according to the manufacturer's 1-step protocol. 24 hr post-transfection the medium was exchanged. 48 hr post-transfection, one tenth of the cells were transferred to 10 cm$^2$ dishes and cultured in medium supplemented with 2 μg/ml puromycin (Sigma-Aldrich). After 5 days of selection, single cells were transferred to 96-well plates using limited dilutions to amplify the clonal lines. Initial screening of these cell lines for edited LRP6 using SDS-PAGE and western blot indicated a CRISPR/Cas9 targeting efficiency of 57%. Genomic DNA from promising candidate cell lines was sent for sequencing (Microsynth AG, Balgach, Switzerland). The results indicated that only heterozygous clones were obtained. siRNA gene knock-down experiments confirmed the heterozygous nature of the cells (*Figure 3—figure supplement 7*). For the CRISPR/Cas9 edited cells used in this work, we confirmed that the fluorescently tagged LRP6 receptors were properly localized at the cell surface (*Figure 3—figure supplement 3b* and *Figure 3—figure supplement 8b*) and underwent

Wnt-dependent post-translational modification similar to wt receptors (*Figure 3—figure supplement 8a*). The CRISPR/Cas9 edited cells were also tested for their ability to activate and inhibit Wnt/β-catenin pathway signaling in response to Wnt and Dkk ligand stimulation, respectively (*Figure 3—figure supplement 2c* and *Figure 3—figure supplement 8a*).

## Cell culture and transient transfection

An 8-well Nunc Lab-Tek chambered cover glass (Thermo Fisher Scientific, Waltham, USA) was coated with fibronectin (Sigma-Aldrich, St. Louis, MO) to ensure efficient adhesion of the cells. To this end, each well was incubated with 200 µl fibronectin (50 µg/ml, diluted in Dulbecco's phosphate buffered saline (DPBS, no calcium, no magnesium, Sigma Aldrich)) for at least 1 hr at room temperature. Afterwards, the solution was removed by aspiration, and the chamber was allowed to dry. Fibronectin coated chambers were either used directly or stored for a maximum of 1 – 2 weeks at 4°C. Before use, the fibronectin-coated wells were washed with DPBS. Stably transfected or CRISPR/Cas9 gene-edited cells were seeded in the pretreated wells; cell experiments were performed on the next day. Wt cells were allowed to attach for at least 1 hr before transient transfection. They were investigated 23 – 36 hr after transfection to ensure comparable results.

Human embryonic kidney 293T (HEK293T) cells (DSMZ ACC-635) were cultured in Dulbecco's Modified Eagle Medium (DMEM, Thermo Fisher, Waltham, MA) supplemented with 10% fetal bovine serum (FBS, Thermo Fisher) and 1% penicillin/streptomycin (P/S; Thermo Fisher). NCI-H1703 human lung carcinoma cells (ATCC CRL-5889) were cultured in RPMI 1640 medium (Thermo Fisher) supplemented with 10% FBS and 1 mM sodium pyruvate (Thermo Fisher). Both cell lines were maintained at 37°C and 5% $CO_2$. Expi293F suspension cells (Thermo Fisher A14527) were cultured in Expi293 Expression Medium (Thermo Fisher) at 37°C and 8% $CO_2$ with 125 rpm orbital shaking. All cell lines tested negative for mycoplasma.

HEK293T wt and NCI-H1703 wt cells were transfected with 1 – 2 µg DNA per well at a 1:6 ratio of MesD:hLRP6-Fluorophore using Xfect (TaKaRa Clontech, Mountain View, CA) according to the manufacturer's protocol. Six hours after transfection, the medium was exchanged with fresh cell culture medium supplemented with 10% FBS and 1 mM sodium pyruvate. For co-expression of LRP6 and xKremen2-eGFP in NCI-H1703 cells, transfections were performed sequentially. First, cells were transfected with 1500 ng LRP6 plasmid and 5 hr later, the same cells were transfected with 400–500 ng xKremen2 plasmid. After 2.5 hr, the medium was completely exchanged and the cells were cultured overnight.

## Preparation of Conditioned Medium (CM)

HEK 293F suspension cells in Expi293 Expression Medium (30 ml, $2.5 \times 10^6$ cells/ml) were transfected with 30 µg *pCS2+human-DKK1/2-eGFP* plasmid using ScreenFectUP-293 + Booster (ScreenFect GmbH) according to the manufacturer's instruction. DKK1-eGFP CM was collected 96 hr post-transfection. TOPFLASH Wnt reporter assays and western blot analysis confirmed both the integrity and activity of the DKK-eGFP fusion proteins produced from suspension cell cultures (*Figure 3—figure supplement 9*). For experiments using NCI-H1703 cells, the harvested cell culture medium was concentrated 30-fold using Amicon 10 kDa ultra filters (Merck KGaA, Darmstadt, Germany) and reconstituted to the same volume with fresh RPMI containing 10% FBS. For experiments with HEK293T cells, the same procedure was followed using DMEM instead of RPMI medium. The CM was sterilized using a 0.22 µm filter and stored at 4°C. DKK2-eGFP CM was prepared using the same protocol but reconstituted to only half the original volume. Mouse Wnt3a CM was prepared from mouse L cells stably transfected with mouse Wnt3a (ATCC CRL-2647). Control CM was prepared from non-transfected L cells (ATCC CRL-2648). Cells were maintained at 37°C and 5% $CO_2$.

## DKK1-eGFP concentration in the CM

The ligand concentration in the CM needs to be measured as precisely as possible because it determines the scaling of the *x*-axis of the binding curves. Thus, any error in this quantity will directly affect the measured $K_D$ values. In this work, we have routinely measured the ligand concentration in CM with solution FCS using PEG-coated sample chambers to minimize errors due to protein adsorption. To examine the reliability of these concentration determinations by independent means, we developed a protocol based on optical absorption and fluorescence spectrometry. A solution of

purified eGFP was prepared to serve as a reference; its concentration was determined with an absorption spectrometer (Cary 100, Agilent Technologies, Santa Clara, CA). Next, the eGFP stock solution was diluted in three steps, each time roughly by a factor of ten, to a final concentration of a few ten nanomolar, similar to the ligand concentration of the CM. To measure the dilution factors as precisely as possible, emission spectra were taken with a Fluorolog-3 spectrofluorometer (HORIBA Jobin Yvon, Edison, NJ) before and after each dilution step (with identical spectrometer settings) to account for pipetting errors and protein loss due to adsorption onto the quartz cuvette surfaces. Then, emission spectra of DKK1-eGFP CM and control CM (without DKK1-eGFP) were taken. Finally, the emission spectrum of DKK1-eGFP CM was quantitatively decomposed into eGFP and background fluorescence spectra by using the spectra of purified eGFP and control CM. This procedure yielded results identical to those from solution FCS.

## siRNA transfection

siRNAs targeting the fluorescently tagged *LRP6* mRNA were designed using the Dharmacon 'siDesign' center. They were obtained from Dharmacon (Horizon Discovery group, Cambridge, UK) and have the following sequences: mCherry, CCUACAACGUCAACAUCAAUU; tdTomato, UCAAAGAG UUCAUGCGCUUUU (MQ-003845–03, siGENOME LRP6 siRNA set of 4 and D-001210–03, siGE-NOME Non-Targeting siRNA #3). siRNA transfection was performed by using ScreenFectA (Screen-Fect GmbH, Eggenstein, Germany) according to the manufacturer's protocol.

## Western blot analysis

Expression of fluorescently tagged proteins was verified by SDS-PAGE/western blot analysis (see *Figure 3—figure supplement 2*, *Figure 3—figure supplement 3*, *Figure 3—figure supplement 4*, *Figure 3—figure supplement 7*, *Figure 3—figure supplement 8*, *Figure 3—figure supplement 9*). For western blot analysis of LRP6 expressed in CRISPR/Cas9 edited, stably transfected and corresponding wt cells, 60 µl control or Wnt3a CM was added to the cells cultured in 96-well plates. After overnight incubation, cells were lysed in 1% Triton lysis buffer (1% Triton X-100, 50 mM Tris-HCl (pH 7.0), 150 mM NaCl, 25 mM NaF, 5 mM $Na_3VO_4$, 0,1% NP-40, 1 mM EDTA) containing Complete protease inhibitor cocktail (Roche, Basel, Switzerland). After 15 min incubation on ice, cell lysates were centrifuged at 10,000 × g at 4°C for 10 min and supernatants were taken.

For membrane protein extraction, adherent cells from confluent 10 $cm^2$ dishes were detached with 2 ml low-salt buffer (5 mM HEPES (pH 7.0), 1 mM $MgCl_2$, 10 mM NaF, 5 mM $Na_3VO_4$) containing Complete protease inhibitor cocktail (Roche, Basel, Switzerland). Detached cells were transferred to a pre-cooled dounce tissue grinder and cell membranes were disrupted using 50 rapid strokes on ice. Lysates were centrifuged at 20,000 × g for 30 min at 4°C to pellet the membranes, which were dissolved in 200 µl 1% Triton lysis buffer.

For western blot analysis of DKK1/2-eGFP, 10 µl of the original, serum-free DKK2-eGFP CM and 0.33 µl of the DKK1-eGFP CM were heat-denatured in Laemmli sample buffer and separated by SDS-PAGE before transfer to a PVDF membrane using a Bio-Rad Transblot-Turbo system (Bio-Rad, Hercules, CA). Membranes were blocked at room temperature for 1 hr in 5% BSA – TBST blocking buffer (5% BSA, 137 mM NaCl, 2,7 mM KCl, 19 mM Tris base (pH 7.4), 0.1% Tween-20) and transferred to a BioLane HTI automated western blot processor for antibody incubation and washing steps. The following antibodies were used: anti-total-LRP6 ([1C10], 1:1000, Abcam, Cambridge, UK), anti-phospho-LRP6 (Sp1490, 1:1000, Cell Signaling, Frankfurt am Main, Germany), anti-mCherry (ab183628, 1:2500, Abcam), anti-GFP (Ab1828,1:2000, Abcam), anti-Vinculin (E1E9V, 1:1000, Cell Signaling), anti-Transferrin receptor (#136800, 1:10,000, Thermo Fisher) and HRP-conjugated anti-rabbit or anti-mouse secondary antibodies (Dako Denmark A/S, Glostrup, Denmark). For semi-quantitative detection of protein bands, the membranes were incubated with ECL Prime (GE-Healthcare, Freiburg, Germany) and imaged using a ChemiDoc Touch Imaging System (Bio-Rad).

## Cell surface biotinylation

To compare the levels of LRP6 localized on the plasma membrane of wt and CRISPR/Cas9 edited NCI-H1703 cells, cell surface biotinylation assays were performed. $1.2 \times 10^6$ cells were seeded in a 6-well plate and, after 24 hr, treated with control CM, DKK1-eGFP CM, or mWnt3a CM for 2 hr. After washing the cells three times with ice-cold PBS (pH 8.0), 1 ml of a freshly prepared biotin

solution (EZ-Link Sulfo-NHS-SS-biotin 5 (Thermo Fisher, Cat. No.21328), 5 mg/ml stock solution in ultrapure water, diluted in PBS to a final concentration of 0.5 mg/ml) was added to each well. The 6-well plate was gently shaken for 1.5 hr on ice. Cells were washed with quenching buffer (35 mM Tris buffer, pH 8.0) to remove un-reacted biotin and washed twice with ice-cold PBS (pH 8.0). Cells were lysed in 300 µl 1% Triton lysis buffer and centrifuged at 10,000 × g at 4°C for 5 min. 30 µl of the resulting supernatant was removed for 'input' analysis. The remaining 250 µl of supernatant was mixed with 50 µl of a 50% NeutrAvidin agarose slurry (Pierce, Cat. No. 29200) and gently shaken overnight at 4°C. Samples were then centrifuged at 500 × g for 1 min and the supernatants collected as 'flow through'. The pelleted beads were washed four times with TBST buffer (Tris-buffered saline with Tween20), then heated to 98°C in Laemmli buffer for 5 min before centrifugation at 10,600 × g for 5 min. Supernatants were collected as' biotin labeled protein' samples. All samples were processed for western blot analysis as described below.

## Luciferase reporter assays

To test the biological activity of the fluorescently tagged LRP6 fusion proteins, $5.5 \times 10^4$ cells cultured in 96-well plates were transfected with 20 ng TCF firefly luciferase (TOPFLASH) and 2 ng CMV Renilla luciferase (*Korinek, 1997*) using ScreenFectA according to the manufacturer's 1-step protocol (ScreenFect GmbH). 12 hr post-transfection, 60 µl of control or Wnt3a CM was carefully added to the cells, which were then incubated for another 6 hr before harvesting cell lysates.

To test the biological activity of the DKK1/2-eGFP fusion proteins, NCI-H1703 lung cancer cells cultured in 96-well plates were transfected with TOPFLASH/Renilla reporter plasmids (20/2 ng). The next day, 80 µl medium was exchanged by either control CM or Wnt3a CM. After 6 hr, 120 µl DKK1/2-eGFP CM was added. The cells were incubated for another 12 hr before harvesting cell lysates.

To determine the biological activity of V5-xKremen2-mCherry/eGFP in HEK293T cells, cells cultivated in 96-well plates were transfected (per well) with 20/2 ng TOPFLASH/Renilla reporter plasmids, 8 ng *mWnt1*, 1 ng *mFrizzled8*, 1 ng *DKK1-eGFP*, and 1 ng *V5-xKremen2-mCherry/eGFP*. Cell lysates were harvested for luciferase reporter assay 24 hr post-transfection.

To compare the responsiveness of CRISPR/Cas9 endogenously tagged cell lines, stably transfected NCI-H1703 cell lines and wt parental cells to mWnt3a and DKK1, cells cultured in 96-well plates were transfected with TOPFLASH/Renilla reporter plasmids (20/2 ng) and, as indicated, 8 ng of *mWnt3a* and 25 ng of *DKK1-eGFP*. Cell lysates were harvested 48 hr post transfection.

For the luciferase assays, cells from 96-well plates were harvested in 35 µl passive lysis buffer (Promega, Mannheim, Germany) and processed according to the manufacturer's protocol. TOPFLASH luciferase values were normalized to control Renilla luciferase values. All error bars shown in the TOPFLASH data (*Figure 3—figure supplement 4*, *Figure 3—figure supplement 7*, *Figure 3—figure supplement 9*) are standard deviations from the mean of 4 samples; experiments were performed at least three times.

## Acknowledgements

We thank Benedikt Prunsche for technical assistance. This work was supported by the Deutsche Forschungsgemeinschaft (DFG) through grant SFB 1324 (projects A6 and Z2). GUN and GD also acknowledge funding by the KIT in the context of the Helmholtz programs Science and Technology of Nanosystems (STN) and BioInterfaces in Technology and Medicine (BIFTM), respectively.

## Additional information

### Funding

| Funder | Grant reference number | Author |
|---|---|---|
| Deutsche Forschungsgemeinschaft | SFB 1324, project A6 | Gerd Ulrich Nienhaus |
| Deutsche Forschungsgemeinschaft | SFB 1324, project Z2 | Gary Davidson |
| Helmholtz-Gemeinschaft | STN | Gerd Ulrich Nienhaus |

| Helmholtz-Gemeinschaft | BIFTM | Gary Davidson |

The funders had no role in study design, data collection and interpretation, or the decision to submit the work for publication.

## Author contributions

Antonia Franziska Eckert, Data curation, Formal analysis, Investigation; Peng Gao, Resources, Data curation, Software, Formal analysis, Validation, Investigation, Visualization, Methodology; Janine Wesslowski, Resources, Formal analysis, Investigation, Visualization; Xianxian Wang, Resources, Formal analysis, Investigation; Jasmijn Rath, Formal analysis, Investigation; Karin Nienhaus, Supervision, Validation, Visualization, Writing - review and editing; Gary Davidson, Conceptualization, Resources, Supervision, Funding acquisition, Validation, Methodology, Project administration; Gerd Ulrich Nienhaus, Conceptualization, Resources, Supervision, Funding acquisition, Validation, Methodology, Writing - original draft, Project administration, Writing - review and editing

## Author ORCIDs

Peng Gao https://orcid.org/0000-0002-5354-3944
Karin Nienhaus https://orcid.org/0000-0002-8357-846X
Gary Davidson http://orcid.org/0000-0002-2264-5518
Gerd Ulrich Nienhaus https://orcid.org/0000-0002-5027-3192

## Decision letter and Author response

Decision letter https://doi.org/10.7554/eLife.55286.sa1
Author response https://doi.org/10.7554/eLife.55286.sa2

# Additional files

## Supplementary files

• Transparent reporting form

## Data availability

All data reported in the paper are included in the manuscript and/or supplementary materials. Source data files have been provided for Figures 2, 5, 6 and 7, and Tables 1 and 2.

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
