## [Decision Letter]

**Acceptance summary:**

Value assessment: Binding of ligands to cellular membrane receptors is fundamental to processes like cell-cell communication, signaling, and initiation of biochemical cascades. Methods that can detect and quantify such interactions are scarce. This "Tools and Resources" contribution introduces a fast variant of line-scanning fluorescence correlation spectroscopy that facilitates detection and quantification of binding of fluorescently modified ligands to their target receptors on cell membranes.

**Decision letter after peer review:**

Thank you for submitting your article "Precise quantification of ligand-cell surface receptor interactions using axial line-scanning FCS" for consideration by *eLife*. Your article has been seen by three peer reviewers, including Hannes Neuweiler as the Reviewing Editor and Reviewer #1, and the evaluation has been overseen by José Faraldo-Gómez as the Senior Editor. The following individual involved in review of your submission has agreed to reveal their identity: Thorsten Wohland (Reviewer #3).

The reviewers have discussed the reviews with one another and the Reviewing Editor has drafted this decision to help you prepare a revised submission.

Major concerns include the novelty of the proposed methodology, estimation of equilibrium dissociation constants, additional controls, and details on data processing and analysis.

Reviewer #1:

The manuscript "Precise quantification of ligand-cell surface receptor interactions using axial line-scanning FCS" by Eckert et al. reports a modification of existing line-scanning fluorescence correlation spectroscopy (lsFCS) for improved detection and quantification of binding strengths of protein-ligand interactions on cell surfaces with live cell applicability. Eckert et al. implement a tuneable acoustic gradient index of refraction (TAG) lens in a multi-colour lsFCS setup that facilitates microsecond scanning of biological samples in axial direction. The axial scan is reported to be two orders of magnitude faster than existing lateral scanning techniques, thus alleviating complications arising from cell membrane dynamics and poor photophysical properties of fluorescent tags. The authors demonstrate the capabilities of their technique through quantitative analysis of inhibition of Wnt signalling using fluorescently modified DKK1 and DKK2 ligands that bind the Wnt co-receptor LRP6 in human lung cancer and human embryonic kidney cells. Eckert et al. test three different methods of LRP6 expression in their assays. Axial lsFCS is finally applied to investigate ternary interactions of DKK1 with the co-receptor Kremen2 and LRP6.

The elucidation of pathways of cellular signalling is a current challenge in molecular biophysics. It requires methods that can detect and quantify protein-ligand interactions in membrane receptors. Temporal analysis of fluorescence fluctuations of modified proteins involving modern multi-colour fluorescence microscopy is a promising approach. Fluorescent background, dynamics of cell membranes and changes in cell morphology, however, complicate a reliable analysis within a cellular environment. The new tool reported by Eckert et al. extends existing lsFCS to overcome such complications by rapidly scanning samples in axial direction thereby improving time resolution in correlation analysis and shortening the overall measurement time. Eckert et al. report clean binding isotherms and the associated equilibrium dissociation constants, measured by their technique, of interactions of DKK1 and DKK2 with the membrane receptors LRP6 and Kremen2. The study involves different fluorescent fusion proteins as tags and different transfection/expression techniques, testifying general applicability. The manuscript is well written. Technique, experimental data together with their analysis are clearly presented and discussed. The advantage of the axial scanning mode in live cell applications appears convincing.

However, the following points should be addressed before publication.

1) The authors report that the Kd of DKK1/LRP6 increases with increasing expression level of LRP6 (subsection “DKK1 and DKK2 binding to LRP6-mCherry”). Eckert et al. rationalize the effect by a reduced affinity of the LRP6 receptor due to packing of LRP6 in the membrane (Discussion paragraph four). But an alternative explanation would be that a significant fraction of DKK1, which is applied at nM concentrations in conditioned medium, binds to over-expressed LRP6 located at cellular sites beyond lsFCS detection. Such undetected binding events can lead to "lost" ligand and would consequently result in artificially high Kds in titration experiments.

2) The authors find similar Kds of DKK1 interactions using mCherry and tdTomato as fluorescent fusions on LRP6 and argue that the change of fluorescent reporters proves reliable and tag-independent measurement of Kds. An important additional control would be to change the tag on the ligand DKK1. eGFP fused to DKK1 may also perturb binding.

3) The authors find that overexpression of unlabelled LRP6 leads to an increase of affinity of DKK1-eGFP. The observation may report on a perturbation of DKK1 binding caused by the fluorescent fusion on LRP6. The possibility of a probe-induced artefact should be discussed.

4) Figure 3—figure supplement 1: the authors detect a decrease of time constant of DKK1-eGFP binding to HEK cells with increasing concentration of DKK1-eGFP. It would be interesting to see if the observed rate constants of binding increase linearly with increasing DKK1-eGFP concentration. This is expected for a bi-molecular reaction in solution. A linear fit to the observed concentration-dependent rate constants yields a microscopic rate constant of DKK1 binding. The microscopic rate constant of ligand-binding to a cellular receptor is interesting because such quantities are usually inferred from in-vitro kinetic experiments on isolated proteins that may not reflect the in-vivo situation.

Reviewer #2:

Nienhaus et al. present a tool for improving fluorescence correlation spectroscopy by axial line scanning using a TAG lens. In large parts the study shows systematic and rigorous experimental work with several proteins interrogated at different densities.

1) Major concern about novelty: The idea of axial scanning to improve the data quality in membrane FCS studies is not new (https://arxiv.org/pdf/1806.00070.pdf) or the work by Jonas Ries (DOI: 10.1039/b718132a). The (only) difference here is that the authors use an ultra-fast TAG lens. Unfortunately, the Introduction is misleading and makes the reader believe that axial scanning FCS is an invention by Nienhaus et al. Moreover, the impact compared to previously reported studies remains unclear.

2) Major concern about data interpretation/processing: One major concern is that it remains unclear how the authors reach the precision reported. How exactly are the data processed and how is the outcome generated? How was the FCS data analyzed? What software program was used? If self-written, why is the source code/ compiled version not provided with example data and manual?

Reviewer #3:

The article by Eckert et al. introduces a new scanning fluorescence correlation spectroscopy (FCS) modality to quantitatively measure molecular affinities of ligand-receptor systems on cell membranes. The authors perform scanning along the optical axis using a tunable lens. This preserves the advantages of lateral scanning, namely the removal of sample movement artefacts, but is much faster as axial scanning by the tunable lens can be performed at higher frequencies than the galvano-scanner based lateral scan systems, reducing measurement time. The authors demonstrate the capability of axial line-scanning FCS by measuring the interaction of various proteins of the Wnt signaling pathway, including DKK1, LRP6, xKremn2, showing that DKK1-LRP6 and Kremen-LRP6 interactions stabilize the trimeric protein complex with a weaker contribution by DKK1-xKremen2 interactions. The article introduces a new method that corrects for many possible artefacts in previous FCS approaches and provides biologically interesting and quantitative Kd values with small errors. The article is well written and convincing but needs to address some concerns:

1) The authors show almost no experimental correlation curves. Especially on cells there is only one set of curves shown in the supplement. But these curves are noisy. At this correlation function quality, how well are the fractions of the different components determined (Equations 8 and 9)? It would also be instructive to see a set of curves at different ligand concentrations.

2) Can the authors either include a positive and negative control?

3) The dissociation constants determined are very strong, below the nanomolar range. Thus the interaction is quite easy to measure and experiments can be performed at low concentrations. Nevertheless, as remarked under point 1 the curves are quite noisy and this will likely get worse at higher concentrations to measure higher Kds. It would be therefore useful if the authors could provide an estimated concentration and Kd range they think they can measure.

4) Subsection “Analysis of fast axial lsFCS on live cells”: The background signal seems to follow the intensity and is not constant as one would assume. Is this an overcorrection or do the authors assume that the background is also bleaching?

5) The authors bleach-correct their time traces. But the correction needs to conserve the variance of the signal, otherwise the amplitudes are changed by the correction. As their correction function is itself a function of time it is not evident whether their transformation fulfills this condition.

In addition, why do the autocorrelations require an extra scaling factor if the intensity traces have already been rescaled to the initial intensity I'(0)? For the cross-correlation they acknowledge that there is no re-scaling necessary. There is no explanation why the cross- should be different from the auto-correlation.

6) The authors state that as the intensities were already corrected, any correction factors would cancel. But overall with bleaching the interacting particles will decrease and that cannot be taken account of in that format. So bleaching will still decrease the cross-correlation amplitude and thus also the determined Kd. Authors might want to discuss this point.

7) The authors use pulsed laser excitation and thus can unambiguously assign photons removing cross-talk. This should result in completely flat cross-correlation functions for a negative control. However, they still have cross-talk from a red label into the green channel when excited with a green dye? This is quite surprising. Thus, a negative control to document this would be very useful.

8) As the background is different for the different wavelength channels, it will lead to different corrections in amplitude for the different correlation functions. This in general will not lead to cancellation effects. While in the authors situation the effect might have been sufficiently small it should not be neglected if axial-line scanning FCS is to be used in general. This point should be elaborated and if the authors think they can neglect background effects, then this should be shown (also note that the shift in Kd is 4 times higher than the error claimed so is not completely negligible).

9) The authors treat non-fluorescent proteins and endogenous proteins the same way. But this is not entirely correct. Cells will have a certain level of endogenous proteins. Depending on the amount of recombinant labeled protein expressed this endogenous protein will have a stronger or weaker influence. If, e.g., 80% of a fluorescent protein are fluorescent, then the measured value of non-fluorescent proteins (parameter β) will be influenced by endogenous proteins. At a high recombinant protein concentration, endogenous proteins play little role and one recovers 80% of fluorescent proteins. But at lower recombinant protein concentrations, the endogenous proteins play a much larger role and values <80% will be recovered for fluorescent proteins. Thus, β is not a constant if one measures on different cells with different recombinant protein expression levels.

10) Increase of Kd with receptor density. This could also be a result of background issues or bleaching, or of the issue with β just mentioned. A positive control at different concentrations would help clarifying the issue.

11) The simplifications constrain the technique. The authors should report how much these assumptions change their actual experimental estimates for the Kd. e.g. the assumption that GFP is 100% fluorescent is not really borne out by recent experiments that rather report 80%. Also the assumption that there are no non-fluorescent tdTomato in the dimers is not entirely justified. Reports on red fluorescent proteins being fluorescent vary between 20 to maximally 60%. Even in the best case this still would lead to 16% non-fluorescent proteins. Furthermore, tdTomato with one or two fluorophores contribute very differently to the correlation functions as the contribution depends not linearly on the brightness. So I agree that the simplifications make the data more easily treatable. But the authors should at least show once the full solution and compare the results of it to the approximation.

---

## [Author Response]

Reviewer #1:The manuscript "Precise quantification of ligand-cell surface receptor interactions using axial line-scanning FCS" by Eckert et al. reports a modification of existing line-scanning fluorescence correlation spectroscopy (lsFCS) for improved detection and quantification of binding strengths of protein-ligand interactions on cell surfaces with live cell applicability. Eckert et al. implement a tuneable acoustic gradient index of refraction (TAG) lens in a multi-colour lsFCS setup that facilitates microsecond scanning of biological samples in axial direction. The axial scan is reported to be two orders of magnitude faster than existing lateral scanning techniques, thus alleviating complications arising from cell membrane dynamics and poor photophysical properties of fluorescent tags. The authors demonstrate the capabilities of their technique through quantitative analysis of inhibition of Wnt signalling using fluorescently modified DKK1 and DKK2 ligands that bind the Wnt co-receptor LRP6 in human lung cancer and human embryonic kidney cells. Eckert et al. test three different methods of LRP6 expression in their assays. Axial lsFCS is finally applied to investigate ternary interactions of DKK1 with the co-receptor Kremen2 and LRP6.The elucidation of pathways of cellular signalling is a current challenge in molecular biophysics. It requires methods that can detect and quantify protein-ligand interactions in membrane receptors. Temporal analysis of fluorescence fluctuations of modified proteins involving modern multi-colour fluorescence microscopy is a promising approach. Fluorescent background, dynamics of cell membranes and changes in cell morphology, however, complicate a reliable analysis within a cellular environment. The new tool reported by Eckert et al. extends existing lsFCS to overcome such complications by rapidly scanning samples in axial direction thereby improving time resolution in correlation analysis and shortening the overall measurement time. Eckert et al. report clean binding isotherms and the associated equilibrium dissociation constants, measured by their technique, of interactions of DKK1 and DKK2 with the membrane receptors LRP6 and Kremen2. The study involves different fluorescent fusion proteins as tags and different transfection/expression techniques, testifying general applicability. The manuscript is well written. Technique, experimental data together with their analysis are clearly presented and discussed. The advantage of the axial scanning mode in live cell applications appears convincing.

We appreciate the overall very positive assessment by Dr. Hannes Neuweiler. In his second paragraph, he summarizes – in impressive clarity – the importance of precise measurements of protein-ligand interactions for biology and identifies the novel and advantageous aspects of our approach. We were delighted about his overly positive summarizing remarks.

1) The authors report that the Kd of DKK1/LRP6 increases with increasing expression level of LRP6 (subsection “DKK1 and DKK2 binding to LRP6-mCherry”). Eckert et al. rationalize the effect by a reduced affinity of the LRP6 receptor due to packing of LRP6 in the membrane (Discussion paragraph four). But an alternative explanation would be that a significant fraction of DKK1, which is applied at nM concentrations in conditioned medium, binds to over-expressed LRP6 located at cellular sites beyond lsFCS detection. Such undetected binding events can lead to "lost" ligand and would consequently result in artificially high Kds in titration experiments.

If we understand the reviewer correctly, he raises the concern that, at nanomolar concentrations (and below), the concentration of free ligands, which governs equilibrium, may be decreased from its nominal value due to receptor binding. Consequently, the more receptors are expressed on the plasma membrane, the greater would be the effect. While this concern is in general justified, we explain why this is not a problem below.

In our experimental procedures, we have taken precautions to avoid this problem. In particular, we have (1) kept cell confluency to <10% surface coverage, (2) used excessive CM volume (500 µL per well) and (3) exchanged the CM in the middle of the incubation period for the lowest ligand concentrations (i.e., <1 nM). See the description in the manuscript.

To explore if there is a problem with free ligand depletion, let us do a rough back-of-the-envelope calculation and estimate the absolute numbers of ligand and receptor molecules: At 1 nM, we have 3 × 10^11^ ligand molecules in 500 µL of CM. The plasma membrane area decorated by receptors is roughly twice the covered area. Thus, at 10% confluency in a 0.8 cm^2^ well, we thus have 0.16 cm^2^ effective area. Therefore, the number of receptors is about 0.16 × 10^8^ µm^2^ × 1000 µm^-2^ (assuming worst case conditions, i.e., the highest receptor density that we encountered), yielding 1.6 × 10^10^ receptors in total per well. Therefore, ligands are still in ~20-fold excess even for this case. By exchanging with fresh medium, we further re-establish the free ligand concentration in the medium.

An additional note: How does ligand depletion manifest itself in binding isotherms such as those reported in Figure 6? The typical signature is that the measured isotherm drops more steeply than expected below a certain ligand concentration because the concentration of free ligands in the CM is no longer maintained at its initial value before adding it to the cells. We have not observed this deviation from the standard Langmuir isotherm behavior even at the highest receptor densities. Therefore, we are confident that this problem does not exist under our experimental conditions.

2) The authors find similar Kds of DKK1 interactions using mCherry and tdTomato as fluorescent fusions on LRP6 and argue that the change of fluorescent reporters proves reliable and tag-independent measurement of Kds. An important additional control would be to change the tag on the ligand DKK1. eGFP fused to DKK1 may also perturb binding.

We agree with the reviewer that careful, quantitative studies of the effect of tags on receptors and ligands are very worthwhile, as fusion tags have the capacity to modify the binding properties to a lesser or greater extent depending on the details of the fusion construct. However, we stress that the topic of our manuscript is to present the axial lsFCS approach; the analysis of effects due to perturbations by the fusion markers is an important yet separate issue.

Notably, our new method appears suitable for careful examination of tag effects because even modest changes in Kd can be revealed. We found, however, that the differences in Kd for LRP6 fused to mCherry and tdTomato are smaller than the reported uncertainties (cmp. Tables 1 and 2). These results appear reasonable in view of the fact that the C-terminally attached fusion tag resides within the cell, whereas DKK binds from the outside. We also believe that the consistent Kd values obtained by independent measurements with different fusion constructs attests to the precision of our method.

To accommodate the reviewer’s request, we have modified the tag on the DKK1 ligand. Instead of using a different fluorescent protein, we have fused an additional SNAP-tag domain (~20 kDa) to the C-terminus of the DKK1-eGFP construct to maintain the fluorescence properties of the tag. Axial lsFCS revealed identical Kds (within the statistical error) and further reinforces our view that the method is robust and precise. See the data in Figure 6, Table 1 and added text in lines paragraph two in subsection “DKK1 and DKK2 binding to LRP6-mCherry”.

3) The authors find that overexpression of unlabelled LRP6 leads to an increase of affinity of DKK1-eGFP. The observation may report on a perturbation of DKK1 binding caused by the fluorescent fusion on LRP6. The possibility of a probe-induced artefact should be discussed.

We were a bit puzzled about this remark because there is no fluorescent fusion on LRP6 in the studies referred to here, in which we analyzed the binding of DKK1-eGFP to xKremen2-mCherry. Thus, we are afraid that he did not grasp the exact point raised here and have rephrased paragraph two of subsection “Interactions between xKremen2-mCherry, DKK1 and LRP6”. Kremen is another receptor involved in the Wnt signaling pathway, and only unlabeled LRP6 was used in these experiments. The mere co-expression of unlabeled LRP6 causes a 20-fold increase in the Kremen-DKK1 binding affinity, which suggests that LRP6 engages in a ternary complex with the other two (fluorescent!) partners. This interpretation is supported by other reports in the literature (e.g., heterodimer formation of Kremen-LRP6), as discussed in the manuscript.

4) Figure 3—figure supplement 1: the authors detect a decrease of time constant of DKK1-eGFP binding to HEK cells with increasing concentration of DKK1-eGFP. It would be interesting to see if the observed rate constants of binding increase linearly with increasing DKK1-eGFP concentration. This is expected for a bi-molecular reaction in solution. A linear fit to the observed concentration-dependent rate constants yields a microscopic rate constant of DKK1 binding. The microscopic rate constant of ligand-binding to a cellular receptor is interesting because such quantities are usually inferred from in-vitro kinetic experiments on isolated proteins that may not reflect the in-vivo situation.

We entirely agree with the reviewer’s point regarding the linear concentration dependence of the apparent on-rate coefficient. We note that modeling of binding with an isotherm, as we do in this work, also requires the appropriateness of a bimolecular reaction. We have to stress, though, that kinetic measurements on single cells are tedious and imprecise due to systematic errors, and do not yield robust kinetic and equilibrium data.

In response to the reviewer’s request, we have added an inset to the figure showing that the apparent rate coefficients are indeed linear in concentration. Moreover, in the caption, we present the on- and off-rate coefficients and the resulting Kds for the single-cell data shown, and also data averaged over an ensemble of 50 cells. Clearly, these painful and time-consuming quantitative imaging experiments are conceptually simple but cannot serve as substitutes for elegant and precise methods such as axial lsFCS.

Reviewer #2:Nienhaus et al. present a tool for improving fluorescence correlation spectroscopy by axial line scanning using a TAG lens. In large parts the study shows systematic and rigorous experimental work with several proteins interrogated at different densities.1) Major concern about novelty: The idea of axial scanning to improve the data quality in membrane FCS studies is not new (https://arxiv.org/pdf/1806.00070.pdf) or the work by Jonas Ries (DOI: 10.1039/b718132a). The (only) difference here is that the authors use a ultra-fast TAG lens. Unfortunately, the Introduction is misleading and makes the reader believe that axial scanning FCS is an invention by Nienhaus et al. Moreover, the impact compared to previously reported studies remains unclear.

We have to admit we were much bewildered by this remark. After thorough consideration, we can only assume that this statement results from a major semantic confusion. In fact, the two references mentioned above do NOT AT ALL address the axial lsFCS method. In the quoted literature, we found that (at least semantically) the most closely related (though still completely different) technique is z-scan FCS. We assume that the reviewer confuses “z-scan FCS” and “axial lsFCS”. To clarify the issue, we briefly explain these very different approaches in the following.

z-scan FCS: By measuring a stack of correlation curves at discrete axial positions (“z-scan”), one can select the correlation curve corresponding to the smallest diffusion time and minimal number of particles. Note that each FCS experiment is performed at a fixed z-position, so there is no z-scan running while FCS data are being taken (despite the notion “z-scan”).

Axial lsFCS: Axial line scanning stands for focus displacement along the optical axis of the microscope, in contrast to lateral line scanning, which describes focus displacement along a line perpendicular to the optical axis, i.e., to the side (“laterally”). Regular, commercial confocal microscopes permit fast lateral line scanning due to the built-in galvanometric scanner. In our manuscript, we give clear credit to previous work introducing lateral line scanning FCS. Displacement along the optical axis is typically accomplished either manually or with a motorized piezo stage changing the distance between the sample and the objective lens. This process is slow due to the inertia involved and, therefore, fast axial line scanning FCS (i.e., on the time scale of 1 ms, which is required for measurements of the kind addressed in our manuscript) was not feasible so far. With the advent of TAG lenses, we are now in the position to scan along the axial direction with periods of several microseconds, as is demonstrated in our manuscript. We emphasize that this is a truly enabling technology and a great advance, and certainly not “the only difference”, as the reviewer states.

Axial scanning offers several advantages, as advertised in the manuscript. First of all, there are many spots on the basal and apical side of a cell membrane to pick for an experiment, which is not the case for lateral scanning. Second, the time resolution is enhanced by more than two orders of magnitude, which facilitates measurement of the full correlation curve (for membrane diffusion). Third, the observation area of axial lsFCS is smaller by a factor of five with respect to lateral lsFCS and spherical, whereas it is elliptical for lateral scanning. The smaller volume is beneficial; it enhances fluctuations and leads to a faster correlation decay, resulting in shorter measurement times to reach the same data quality (see Figure 2D, E). This, in turn, leads to less photobleaching and suppresses the probability of perturbations during the experiment, e.g., by strongly fluorescent vesicles moving close to the plasma membrane. This latter point will be much appreciated by researchers familiar with FCS on live specimens. Fourth, there is no anisotropy parameter included in the fit, so the results are more robust. Finally, for dual-color cross-correlation experiments, ensuring focal volume/area overlap is much easier with axial line scanning because of “axial self-alignment”. By contrast, lateral lsFCS or 3D FCS cross-correlation analysis require careful minimization of axial displacements of the observation volumes, e.g., due to chromatic aberrations of the objective lenses.

2) Major concern about data interpretation/processing: One major concern is that it remains unclear how the authors reach the precision reported. How exactly are the data processed and how is the outcome generated? How was the FCS data analyzed? What software program was used? If self-written, why is the source code/ compiled version not provided with example data and manual?

In the original manuscript, we have aimed to describe the data analysis in a detailed fashion to make the general procedures comprehensible to the general reader. It was, however, not our intention to publish an FCS software package. We have been involved in FCS-based research for more than 20 years and have always relied on writing our own code for the analysis of FCS data for more than two dozen papers using this method. Here, we have employed our own MATLAB code for intensity time trace analysis including background correction, to calculate correlation functions and to fit the correlation data with the model functions. These MATLAB routines are strongly interwoven with the specific microscopy hardware employed. The fitted parameters of the correlation functions were then exported to OriginPro for analysis of the concentration dependence of the function Γ (*C*_L_), Equation 5. Therefore, the errors quoted for the Kds are statistical errors provided by commercial software.

To further clarify the analysis procedures, in the revised manuscript, we have rewritten the Materials and methods part addressing the data analysis workflow, moved Figure 2—figure supplement 2 to the main text (now Figure 4) and included a flowchart (Figure 4—figure supplement 3) that visualizes the individual steps taken to determine binding isotherms and Kds.

Reviewer #3:The article by Eckert et al. introduces a new scanning fluorescence correlation spectroscopy (FCS) modality to quantitatively measure molecular affinities of ligand-receptor systems on cell membranes. The authors perform scanning along the optical axis using a tunable lens. This preserves the advantages of lateral scanning, namely the removal of sample movement artefacts, but is much faster as axial scanning by the tunable lens can be performed at higher frequencies than the galvano-scanner based lateral scan systems, reducing measurement time. The authors demonstrate the capability of axial line-scanning FCS by measuring the interaction of various proteins of the Wnt signaling pathway, including DKK1, LRP6, xKremn2, showing that DKK1-LRP6 and Kremen-LRP6 interactions stabilize the trimeric protein complex with a weaker contribution by DKK1-xKremen2 interactions. The article introduces a new method that corrects for many possible artefacts in previous FCS approaches and provides biologically interesting and quantitative Kd values with small errors. The article is well written and convincing but needs to address some concerns:

We thank reviewer #3, Prof. Thorsten Wohland, for his clear summary of our work pointing to the advantages of the technique, and his overall positive assessment. He has evaluated the manuscript in great detail and raised a number of concerns that we have addressed below.

1) The authors show almost no experimental correlation curves. Especially on cells there is only one set of curves shown in the supplement. But these curves are noisy. At this correlation function quality, how well are the fractions of the different components determined (Equations 8 and 9)? It would also be instructive to see a set of curves at different ligand concentrations.

We are happy to follow the reviewer’s suggestion to present more experimental correlation curves. For illustration, we have included two sets of example correlation curves, one with variation of receptor density and one with variation the DKK ligand concentration, as Figure 4—figure supplement 1.

Indeed, our correlation curves are noisier than what is often presented because we have limited our data acquisition time to 60 s. Obviously, we could measure longer to achieve better time averaging. Then, however, photobleaching would be more severe and require more extensive corrections. Although our bleaching correction is mathematically strictly correct, the overall signal-to-background ratio will be worse for longer traces, causing greater uncertainties in the Kd determination in the end. Even though the pair correlations show significant noise, a fit with only two free parameters (Equation 1) gives robust results. We note that the specific advantages of axial over lateral lsFCS are beneficial for the extraction of parameters, (1) the steeper temporal decay of the correlations and (2) the simpler model function with only two rather than three fit parameters. For better statistics, we take multiple measurements on several cells at each ligand concentration. In the next step, we determine *K*_D_ from the ligand concentration dependence of the function Γ(*C_L_*) shown in Equation 5. In this procedure, we fit experimental data comprising several ten measurements with only two varying parameters. Thus, the problem is mathematically overdetermined, resulting in a high precision of K_D_.

2) Can the authors either include a positive and negative control?

We fully agree with the reviewer that it is worthwhile to frequently carry out positive and negative control experiments to ensure proper functioning of the apparatus and data analysis routines. Thus, in response to the reviewer’s request, we have included three sets of control experiments on live HEK293T cells as Figure 4—figure supplement 2 (described in subsection “Analysis of fast axial lsFCS on live cells”). As a negative control, we have co-expressed our LRP6-mCherry receptor with the membrane marker Mem-eGFP. Both molecules independently diffuse within the plasma membrane but do not interact. Accordingly, the resulting cross-correlation has zero amplitude. As a positive control, we have used LRP6-mCherry-eGFP, so that each receptor carries a green and a red fluorescent (or non-fluorescent) protein. As another positive control, we have used LRP6-tdTomato, which can be excited both with 470 and 561 nm lasers (PIE). This experiment is expected to yield complete cross-correlation for the two color channels, so that the correlation amplitudes differ only due to the wavelength-dependent observation areas. Experimental details have been included in the figure caption.

3) The dissociation constants determined are very strong, below the nanomolar range. Thus the interaction is quite easy to measure and experiments can be performed at low concentrations. Nevertheless, as remarked under point 1 the curves are quite noisy and this will likely get worse at higher concentrations to measure higher Kds. It would be therefore useful if the authors could provide an estimated concentration and Kd range they think they can measure.

First of all, we would like to point out that affinities/avidities of biological receptors toward their cognate ligands are typically high (subnanomolar to nanomolar) to allow specific binding even under conditions of low abundance of ligand. Thus, our method is not limited to receptor-ligand binding in the Wnt signaling pathway but should be broadly applicable to cell signaling interactions in general. We do not agree, though, with the reviewer that it is quite easy to measure at low concentrations. Quantitative studies at subnanomolar concentrations have their own experimental challenges because equilibration times tend to become very long and the cells need to stay in good shape during long equilibration times. Moreover, molecule numbers and thus fluorescence signals are small while autofluorescence remains high, and the ratio of ligand versus receptor numbers is no longer large. There is also the risk that surface adhesion causes a significant concentration decrease when handling picomolar solutions (CM) unless great care is being taken to avoid such problems.

The reviewer is perfectly correct, however, in stating that there are intrinsic limitations of the method for studies of systems with larger Kd (lower affinity). Measurements of binding isotherms requires higher ligand concentrations, fluctuation amplitudes decrease while, at the same time, there is much higher background in the extracellular space filled with CM and, after a while, also in the cytosol due to endocytosis. As we have described in the manuscript, we extract fluorescence intensities from the membrane as well as background near the membrane from our axial lsFCS data. At high ligand concentrations, background may become so large that it dwarfs the signal. To arrive at a reasonable estimate, we may take a signal-to-background ratio of 1:10 as the limiting case. We note that this background is taken to be uncorrelated and just represents an offset in our intensity time traces. The extracellular volume contributing to the background of the membrane signal is half the confocal volume, i.e., 0.1 fL (assuming *ω*_0_ = 207 nm, z_0_ = 5*ω*_0_). Thus the average number of ligands causing fluorescent background from this volume is 0.06 *C_L_* [nM]. The number of receptors is simply the area density times the observation area (0.14 µm^2^). In a “worst case” scenario, i.e., for only 20 receptors per squared micron (as in our CRISPR cell line), we have ~3 receptors within the observation area on average. Assuming equal brightnesses of receptors and ligands, we can thus work with up to 3/0.06 = 50 nM ligand concentration. This is a conservative estimate; see Figure 7B for data with 220 nM ligand concentration. If we want to be very thorough with our Kd determination and aim to measure the binding isotherm within +/– 1 order of magnitude around Kd (a rule that is often seen not to be obeyed), the upper value of Kd would be 5 nM (for low-level receptor expression). This value scales linearly with the receptor density and, thus, Kds up to 50 nM can be measured with 200 receptors per micrometer squared, a typical value for transiently transfected, overexpressing cells. If we relax the requirement of measuring the full binding curve (10% – 90%), we may gain up to an order of magnitude. Finally, at even higher concentrations, we entirely lose our intensity fluctuations on which FCS is based. We have included the results of these considerations in subsection “Interactions between xKremen2-mCherry, DKK1 and LRP6”.

4) Subsection “Analysis of fast axial lsFCS on live cells”: The background signal seems to follow the intensity and is not constant as one would assume. Is this an overcorrection or do the authors assume that the background is also bleaching?

Indeed, as the reviewer recognizes, the background shown in Figure 4 is not constant. Bleaching of background from the extracellular side can be excluded but may in fact occur on the intracellur side of the plasma membrane (static or slowly moving cellular structures). However, this apparent bleaching of the background mainly results from the fact that a fraction of the membrane fluorescence emitted by the periphery of the focus was included in the background. Subtraction of a small fraction of the signal included in the background effectively rescales the intensity trace and will not influence the correlation results. We prefer to lose some signal in this way instead of measuring the background further away from the membrane location, which introduces additional uncertainties.

5) The authors bleach-correct their time traces. But the correction needs to conserve the variance of the signal, otherwise the amplitudes are changed by the correction. As their correction function is itself a function of time it is not evident whether their transformation fulfills this condition.

Here we disagree with the reviewer. Indeed, we correct the intensity time traces to restore a constant mean intensity, which evidently does not conserve the variance. However, there is no requirement to conserve the variance of the intensity time trace. Instead, we can later apply this correction to the intensity-normalized variance of the (raw) autocorrelation function, *G*(0), which is used for our further analysis. This correction is introduced in Equation 3 and, as this procedure is not evident to the reviewer, we sketch the main ideas in the following.

We start with the key assumption that the bleaching effect is quasi-stationary over the correlation time window. Otherwise, photobleaching would distort the correlation function itself. This approximation holds very well, given that the correlations decay within a fraction of a second (thanks to the axial scanning geometry, see Figure 2D) and the bleaching function *γ*(*t*) varies slowly over the total duration of the trace (60 s). Then, the overall autocorrelation function, *G*(τ), of the entire trace can be computed as an average over all sub-correlation functions, *G*_i_(τ), taken over short time intervals, *i*, along the trace, with *γ*(*t*) = *γι* constant. As explained in the manuscript, we use *I*(*t*) = *I*’(*t*) *γ*(*t*), where the prime denotes raw (uncorrected) data. As usual, *I*’(*t*) is assumed to be strictly proportional to the number of fluorophores in the sensitive volume. For each sub-correlation, *G*_i_(0) = 1 / <*I*’(0)>_i_ = *γ*_i_ / <*I*(0)>_i_ (Poisson process); averages (< >) are calculated over the time interval *i*. With increasing time (index *i*), the correlation amplitude increases as a consequence of fluorophore depletion. We would like to calculate the amplitude at the earliest times of the trace, *G*_0_(0), because that condition reflects the fluorophore numbers or concentrations in the absence of photobleaching. Thus, with *γ*_0_=1,*G*_0_(0) = <*I*’(0)>_0_^-1^ = <*I*(0)>_0_^-1^. If we then average over all *G*_i_(0) to calculate the *G*(0) of the entire trace, we simply obtain an additional factor of <*γ*> (averaged over all intervals) in the numerator, which we have to cancel to arrive at the expression for *G*_0_(0), as is done in Equation 3.

In addition, why do the autocorrelations require an extra scaling factor if the intensity traces have already been rescaled to the initial intensity I'(0)? For the cross-correlation they acknowledge that there is no re-scaling necessary. There is no explanation why the cross- should be different from the auto-correlation.

We are a bit puzzled about the first question because, in his previous point, reviewer 3 rightfully argues that the mean but not the variance has been corrected in the intensity time trace. Please see our response to the previous point for the explanation of the extra scaling factor of the autocorrelation function.

For the cross-correlation, there is no correction required according to the following line of argument. The amplitude *G*_×_(0) is given by *N*_gr_ / [(*N*_r_ + *N*_gr_)(*N*_g_ + *N*_gr_)], with the numbers of complexes, receptors and ligands denoted *N*_gr_, *N*_r_ and *N*_g_. Let us focus on photobleaching of (red) receptors: Bleaching depletes ligand-receptor complexes and monomeric receptors equally in terms of relative fractions (not absolute numbers!). If we find 10% bleaching of receptors engaged in complexes, we also expect 10% bleaching of monomeric receptors (as long as binding does not affect the photophysical properties). Thus, the bleaching effect will cancel in the expression for *G*_×_(0) shown above because it affects the numerator and denominator in the same fashion. Moreover, the law of mass action, which holds for a bimolecular reaction in dynamic equilibrium, will restore the monomer-complex ratio even if there were an imbalance (within the characteristic time of the reaction). For the bleaching of (green) ligands, the same argument holds (for symmetry reasons). Note that *N*_g_ = 0 because ligand diffusion in the solvent is too fast to cause fluctuations in our time window, and there are no slowly moving ligands on the plasma membrane unless they are specifically bound to the receptors.

6) The authors state that as the intensities were already corrected, any correction factors would cancel. But overall with bleaching the interacting particles will decrease and that cannot be taken account of in that format. So bleaching will still decrease the cross-correlation amplitude and thus also the determined Kd. Authors might want to discuss this point.

We are afraid that we disagree with the reviewer on this point and refer to our response to his point #5 for the explanation as to why the cross-correlation is not affected by bleaching. The reviewer is certainly correct in stating that bleaching decreases the number of interacting particles, which are in the numerator in the *G*_×_(0) expression given in our response to the previous point. However, the concomitant decrease of the denominator ensures that the cross-correlation amplitude retains its magnitude. You can also look at Equation 4 and scale the intensities with a constant <1 to mimic photobleaching, which can be pulled out from the averages in the numerator and denominator. This again requires that we assume quasi-stationarity of bleaching effects (weak bleaching) over the time window of correlation.

7) The authors use pulsed laser excitation and thus can unambiguously assign photons removing cross-talk. This should result in completely flat cross-correlation functions for a negative control. However, they still have cross-talk from a red label into the green channel when excited with a green dye? This is quite surprising. Thus, a negative control to document this would be very useful.

We are most thankful to the reviewer for drawing our attention to this issue. He is completely correct that, under ideal circumstances, a negative control such as a receptor-expressing cell in medium without DKK ligands should give zero cross-correlation amplitude. Nevertheless, in the data collected, we frequently found finite cross-correlations (which is in the numerator of *Γ*(*C*_L_)) even without any ligands in the solution.

We have looked into this issue and made additional experiments to gain a thorough understanding of the problem. First, we performed careful in-vitro spectroscopic analyses on mCherry, showing that the fraction excited with 470 nm light and emitted into the green channel should only be 0.16% with respect to the red emission and not 4 – 5% as quoted in the original manuscript. Thus, crosstalk by red-fluorescent mCherry is NOT a viable explanation. Another possibility could be the presence of a green-fluorescent mCherry byproduct which is a well known dead-end intermediate in the fluorophore maturation pathway. However, we have convinced ourselves that this cannot contribute in a sufficiently strong way. The real problem became clear when we scrutinized our raw data and noticed that a small amount of correlated background occurred in some of the measurements whenever there were bright intracellular fluorescent structures in the vicinity of the plasma membrane close to the point chosen for axial scanning. The wavelength band of the green detection channel [500 – 550 nm] agrees with the wavelength region where cellular autofluorescence predominantly emits (Aubin, J. Histochem. Cytochem. 27: 36-43 (1979)). We suspect that this correlated background comes from moving vesicles close to the membrane. Indeed, longer correlation times than those of single receptors were typically found in these cases. This background problem becomes more severe for cells expressing receptors at a higher level, which justifies in a way that we treated this component proportionally to the receptor density in the derivation of Equation 5.

To solve this problem, we implemented a stringent data quality management that removes problematic correlation data from the analysis. The rejection criteria are described in Figure 4—figure supplement 3. Without this crosstalk complication, the Kd analysis becomes significantly simpler, see Γ(*C*_L_) in the revised Equation 5. Accordingly, we have rewritten the entire Materials and methods part referring to this issue, and we have also added a derivation for the case that maturation of the label attached to the ligand is not complete. Finally, while the analysis without included crosstalk leads to a considerably changed and in fact simplified expression for Γ(*C*_L_)) due to the absence of correction factors, we emphasize that, because the binding isotherm governs the expression, there are only small changes of the resulting Kd values in comparison to the earlier version.

8) As the background is different for the different wavelength channels, it will lead to different corrections in amplitude for the different correlation functions. This in general will not lead to cancellation effects. While in the authors situation the effect might have been sufficiently small it should not be neglected if axial-line scanning FCS is to be used in general. This point should be elaborated and if the authors think they can neglect background effects, then this should be shown (also note that the shift in Kd is 4 times higher than the error claimed so is not completely negligible).

We believe that the reviewer may have misunderstood our procedures. Background is removed from ALL intensity time traces in our standard analysis and, therefore, this is not a relevant issue for our approach to determine Kd, so there is really no need to cancel anything.

However, in Figure 5, we demonstrate that we can use the time traces even without background subtraction and, nevertheless, we end up with only a slightly changed Kd. This exercise was only included to illustrate that background effects cancel to a high degree in the function Γ(*C*_L_). We find the reviewer’s criticism that “the shift in Kd is 4 times higher than the error claimed so is not completely negligible” somewhat harsh because the error reported is solely of statistical nature, whereas Kd values reported in the literature typically suffer from (often much larger) systematic errors that are hard to estimate. For example, it is not at all trivial to generate CM solutions with precisely defined subnanomolar concentrations of ligands and to maintain that concentration when immersing the cells.

We still want to comment on the observation that the (uncorrelated) background cancels to a great extent. We can write the intensity time trace as the sum of signal and background,(1)Ic(t)=Icsig(t)+Icbg(t).

Here, the index *c* = *R*, *G* stands for the red and green color channels, respectively. Then, according to Weidemann et al., 2002, the correlation functions are(2)Gc(τ)= 1(1+ηc)2 Gc0(τ),(3)G×(τ)= 1(1+ηr)(1+ηg) G×0(τ),with ηc=Icbg(t)/Icsig(t). The superscript “0” on the right hand sides of Equations 2 and 3 denotes correlation functions in the absence of background. Obviously, the factors in front of these correlation functions cancel in the expression *G*_x_^2^(0)/[*G*_R_(0)*G*_G_(0)], Equation 5 in the manuscript. Interestingly, we also came across a nicely written book chapter (Foo et al., 2012a) from the reviewer’s lab. Just look at your own Equations 26 and 27. In *G*_x_^2^(0)/[*G*_R_(0)*G*_G_(0)], all prefactors correcting for background modification of the amplitudes cancel.

9) The authors treat non-fluorescent proteins and endogenous proteins the same way. But this is not entirely correct. Cells will have a certain level of endogenous proteins. Depending on the amount of recombinant labeled protein expressed this endogenous protein will have a stronger or weaker influence. If, e.g., 80% of a fluorescent protein are fluorescent, then the measured value of non-fluorescent proteins (parameter β) will be influenced by endogenous proteins. At a high recombinant protein concentration, endogenous proteins play little role and one recovers 80% of fluorescent proteins. But at lower recombinant protein concentrations, the endogenous proteins play a much larger role and values <80% will be recovered for fluorescent proteins. Thus, β is not a constant if one measures on different cells with different recombinant protein expression levels.

We completely agree with the reviewer that the dark receptor fraction includes both non-matured, “exogenous” fusion proteins and endogenously produced receptors. Accordingly, in the original manuscript, we wrote: “On the one hand, *β* depends on the fraction of endogenously produced (unlabeled) receptors, which are present also in the (haploid) gene edited cells; on the other hand, it depends on the ability of the gene construct and the cell line to produce fully mature, fluorescent fusion proteins.” This sentence has not been changed in the revised version.

Endogenous receptors play a role if and only if the cells produce them in quantities that are not negligible with respect to the exogenous proteins. From the reported receptor densities (Figure 6), it is clear that endogenous proteins are an absolute minority species in overexpressing cells. Note that the *β* values in the Tables reflect the behavior described by the reviewer. We emphasize that this parameter affects only the amplitude of Γ(*C*_L_)) and thus has no influence whatsoever on Kd.

In the original manuscript text, we defined *β*as the fraction of receptors carrying a functional fluorescent domain, implying that the fraction is calculated with respect to all receptors that can bind the cognate ligand. To state this fact explicitly, we have now modified the sentence to read: “…the ratio between receptors carrying a functional fluorescent domain and all (exogenous or endogenous) receptors capable of specific binding.”

10) Increase of Kd with receptor density. This could also be a result of background issues or bleaching, or of the issue with β just mentioned. A positive control at different concentrations would help clarifying the issue.

In our responses to previous points of reviewer 3, we have shown that there are no background or bleaching issues or issues with the parameter *β* that we have reasons to worry about. To avoid misunderstandings, we again briefly reiterate these points.

Parameter *β*:Τhere is no issue at all. In our response to point #9, we have stated that this parameter obviously depends on the amount of endogenously produced receptors, as we have never disputed, the reviewer has recognized and our data also indicate. A look at our model equation (Equation 5) will convince the reader that *β* is merely a scaling factor of the binding isotherm and does not affect Kd at all.

Bleaching issues: As we report in the manuscript, the average bleaching factor <*γ*> is small in all our measurements (only ~1.5, i.e., ~30% bleaching), thanks in part to short data acquisition times, and well compensated by following a strict mathematical procedure (Equation 3). Thus, it makes no sense to suspect that the Kd variation with receptor density (Figure 6) can be due to bleaching issues.

Background issues: As we have stated in the manuscript and our responses above, background is subtracted from the membrane-derived intensities and thus has no role to play here. This argument is further strengthened by our response to point #8, namely, that background effects are largely compensated by our analysis strategy; even without any correction, the Kd value is minimally affected (Figure 5). Moreover, we can infer from the data themselves that the shift of Kd is not due to background issues. In our experiments on NCI-H1703 cells (Figure 6, Table 1), the background-to-signal ratio increased in the order transient transfection (receptor density (median): 139±82 μm^2^) stable transfection (density: 45±23 μm^2^) CRISPR/CAS9 (density: 18±9 μm^2^) and thus as expected from the decrease in receptor numbers. Despite the difference in receptor density by more than twofold, there is no shift of Kd between the latter two samples, however. Only for densities above the one observed for stably transfected cells do we observe the shifts, suggesting that there is a critical density above which the effect appears. A further piece of evidence that the reported small changes of Kd should be taken seriously is the comparison of DKK1 and DKK2. We determined *K*_D_ = 0.26 ± 0.04 nM and 0.08 ± 0.01 nM for Dkk2-LRP6 and DKK1-LRP6 binding in stably transfected cells, respectively, and thus a significantly lower affinity of LRP6 toward DKK2 under similar expression conditions. This finding is in perfect agreement with literature values from in-vitro studies (Mao et al., 2001).

11) The simplifications constrain the technique. The authors should report how much these assumptions change their actual experimental estimates for the Kd. E.g. the assumption that GFP is 100% fluorescent is not really borne out by recent experiments that rather report 80%. Also the assumption that there are no non-fluorescent tdTomato in the dimers is not entirely justified. Reports on red fluorescent proteins being fluorescent vary between 20 to maximally 60%. Even in the best case this still would lead to 16% non-fluorescent proteins. Furthermore, tdTomato with one or two fluorophores contribute very differently to the correlation functions as the contribution depends not linearly on the brightness. So I agree that the simplifications make the data more easily treatable. But the authors should at least show once the full solution and compare the results of it to the approximation.

We were a bit puzzled by this remark. The reviewer refers to a part of the manuscript that addresses the analysis of fractions of heterodimeric receptor-receptor complexes and not Kd determination. Therefore, it does not appear meaningful to request that we “….should report how much these assumptions change their actual experimental estimates for the Kd.” Moreover, this part concerns a small side issue disconnected from the main topic, which is Kd determination. Here we wanted to show that our data give evidence of heterodimeric xKremen2-LRP6 complexes in the plasma membrane. We did not aim to be strictly quantitative; this is why we introduced a number of simplifications which, however, do not have any implication for our conclusion that xKremen2-LRP6 heterodimers exist.

The reviewer asked us to discuss the validity of our assumptions and compare the results with the full solution. However, since this is only a side issue, we feel that it does not make sense to expand this part of the Materials and methods section even further. Thus, we have instead aimed to shorten this subsection while, at the same time, accommodating the reviewer’s requests to avoid simplifications. To this end, we have rewritten this entire subsection to treat the maturation yields of the fluorescent proteins explicitly in the equations. Because the maturation yield is not a precise, characteristic parameter of a chosen type of fluorescent marker protein but depends on the specifics of the experiment, we assume values of maturation yields that appear reasonable to us (and hopefully also to readers) and show the extents of the resulting corrections.